# Air quality related equity implications of U.S. decarbonization policy

Paul Picciano[1,7], Minghao Qiu [2,3,7], Sebastian D. Eastham [4,5], Mei Yuan[5], John Reilly [5] & Noelle E. Selin [1,6] ✉

Climate policies that target greenhouse gas emissions can improve air quality by reducing co-emitted air pollutant emissions. However, the extent to which climate policy could contribute to the targets of reducing existing pollution disparities across different populations remains largely unknown. We quantify potential air pollution exposure reductions under U.S. federal carbon policy, considering implications of resulting health benefits for exposure disparities across U.S. racial/ethnic groups. We focus on policy cases that achieve reductions of 40-60% in 2030 economy-wide carbon dioxide ($CO_2$) emissions, when compared with 2005 emissions. The 50% $CO_2$ reduction policy case reduces average fine particulate matter ($PM_{2.5}$) exposure across racial/ethnic groups, with greatest benefit for non-Hispanic Black ($-0.44\ \mu g/m^3$) and white populations ($-0.37\ \mu g/m^3$). The average exposure disparity for racial/ethnic minorities rises from 12.4% to 13.1%. Applying an optimization approach to multiple emissions reduction scenarios, we find that no alternate combination of reductions from different $CO_2$ sources would substantially mitigate exposure disparities. Results suggest that $CO_2$-based strategies for this range of reductions are insufficient for fully mitigating $PM_{2.5}$ exposure disparities between white and racial/ethnic minority populations; addressing disparities may require larger-scale structural changes.

Emissions of greenhouse gases (GHG) that contribute to climate change are often associated with air pollutant emissions that lead to formation of fine particulate matter ($PM_{2.5}$). $PM_{2.5}$ causes upwards of ~200,000 premature deaths in the U.S. annually and disproportionally harms U.S. racial/ethnic minorities and low-income populations[1,2]. A growing body of literature has demonstrated how policies aiming to reduce GHG emissions can concurrently reduce air pollution and improve public health[3]. However, there remains disagreement on both the direction and magnitude of effects of such policies on disparities in exposure. Addressing disparities in air pollution exposure to racial/

ethnic minorities and low-income populations and mitigating climate change risks are both closely tied to existing policy goals: In January 2021, the US government announced a target that 40% of the overall benefits of certain federal investments, including investments in the areas of clean energy, should flow to disadvantaged communities (defined as those underserved and overburdened by pollution)[4].

Air pollution exposure disparities have persisted despite improvements in air quality[5–7]. Disparities by race/ethnicity are greater than disparities by income and exist across all income groups[1,6]. Tessum et al. estimated that in 2014, Black and Hispanic populations were

[1]Institute for Data, Systems, and Society, Massachusetts Institute of Technology, Cambridge, MA 02139, USA. [2]Doerr School of Sustainability, Stanford University, Stanford, CA 94305, USA. [3]Center for Innovation in Global Health, Stanford University, Stanford, CA 94305, USA. [4]Laboratory for Aviation and the Environment, Department of Aeronautics and Astronautics, Massachusetts Institute of Technology, Cambridge, MA 02139, USA. [5]Joint Program on the Science and Policy of Global Change, Massachusetts Institute of Technology, Cambridge, MA 02139, USA. [6]Department of Earth, Atmospheric and Planetary Sciences, Massachusetts Institute of Technology, Cambridge, MA 02139, USA. [7]These authors contributed equally: Paul Picciano, Minghao Qiu. ✉e-mail: selin@mit.edu

exposed to 56% and 63% more $PM_{2.5}$ than they were responsible for based on consumption; in contrast, the non-Hispanic white population experienced 17% less[8]. Another study showed that most sources of $PM_{2.5}$ disproportionately harm racial/ethnic minorities, except for coal-fired electricity generation and agriculture[1]. These disparities in part reflect systemic environmental racism, including the long-lasting consequences of discriminatory practices such as redlining[9].

Many studies have evaluated air quality related health benefits of climate and clean energy policies (sometimes referred to as "co-benefits"). Gallagher and Holloway review 26 such studies[3], including several that found that monetized air pollution related health benefits can exceed the estimated climate benefits as well as implementation costs of the policy alone[10,11]. While the impact of carbon reductions is the same regardless of the location of emissions, the local nature of $PM_{2.5}$ exposure means that changes in air pollution-related health burdens due to policy can be unequally distributed. Communities affected by polluting sources with lower marginal abatement costs (i.e. the cost of reducing one unit of $CO_2$ emissions from the sources) will typically benefit more from policies that involve carbon pricing[12,13]. Furthermore, reductions in one location may result in increased emissions outside of the policy coverage ("leakage") that could increase exposures[11]. This has led some to argue that market-based carbon policies will not address air pollution disparities, leading to efforts such as in California and Washington to adopt distinct and explicit equity-related provisions as complements to carbon pricing[14].

Much existing research evaluating air pollution equity impacts of climate policy has focused on retrospective analyses of existing policies, largely in California, finding limited but mixed effects on equity outcomes. For example, Cushing et al. estimate that California's 2013 GHG cap-and-trade program exacerbated inequities, finding that over half of covered facilities increased emissions (with total emissions remaining under the cap) and that areas within 2.5 miles of facilities with increased emissions had higher shares of racial/ethnic minorities and low-income populations than areas with decreased emissions[15]. In contrast, Anderson et al. find limited equity impacts of the same program by comparing changes in emissions for disadvantaged counties[16]. Hernandez-Cortez and Meng apply an atmospheric dispersion model to track transport of primary pollutants as well as a reduced-form chemical transport model including secondary $PM_{2.5}$ formation, finding that while disparities had been increasing before the cap-and-trade program, the program reduced disparities but did not eliminate them[13]. Focusing on the continental US, Qiu et al. find that the historical deployment of wind power has heterogeneous impacts of air pollution exposure disparities, i.e., widening the disparity gap in some states while narrowing the gap in other states[17].

A few studies have considered equity impacts of future decarbonization scenarios, mostly focusing on selected regions or specific policies. Li et al., focusing on California, apply an energy-economic optimization model and a chemical transport model (CTM) to evaluate low carbon energy scenarios in 2050, finding that reducing GHG emissions by 80% relative to 1990 levels could reduce racial/ethnic $PM_{2.5}$ disparities by up to 20%[18]. Zhu et al. find in a study of California that the magnitude and distribution of health benefits varies among scenarios reducing economy-wide GHG emissions by 80%[19]. Luo et al. for Texas, found that power sector decarbonization there yields health benefits but fails to address air pollution inequities[20]. The report by Diana et al. constructs a national policy scenario that reduces $CO_2$ emissions by 20% and air pollution damages by 50% for Black, Hispanic, and low-income populations specifically[21]. Polonik et al. recently examined a range of U.S. GHG reduction scenarios, finding that least-cost and income based reductions can exacerbate air pollution disparities for racial/ethnic minorities, and that reducing transportation emissions has the most potential to reduce racial inequities[22].

Here, we examine the underlying fundamental question of whether and to what extent national $CO_2$ policy with an ambition level comparable to near-term federal goals can mitigate racial/ethnic disparities in air pollution exposure. We use energy-economic scenarios and an air quality model to quantify whether and how different policies that reduce carbon dioxide ($CO_2$) emissions by 40–60% in 2030 relative to 2005 levels simultaneously reduce racial and ethnic air pollution disparities at national scale. A 50% reduction by 2030 is consistent with the U.S. pledge under the Paris Agreement (which aims to reduce emissions by 50–52% compared to the 2005 level) and the 2022 Inflation Reduction Act (IRA)[23]. We evaluate the extent to which carbon policies of comparable magnitude and sectoral scope can feasibly achieve reductions in air pollution disparities.

In contrast to studies focusing on selected regions or specific policy designs, our economy-wide approach allows us to identify the national-scale implications and trade-offs of carbon reduction strategies. We focus on 40–60% reductions in economy-wide emissions by the year 2030, when compared with 2005 levels. We apply modeled energy-economic scenarios of a cap-and-trade program to estimate policy-induced emissions reductions for a primary policy case of 50% reductions, and use a reduced-form air quality model to evaluate $PM_{2.5}$-related equity outcomes including impacts of disparities in exposure over the continental US. We then quantify the degree to which alternative distributions of $CO_2$ emissions reductions (with the same level of aggregated emissions reductions, and for two additional policy cases of 40% and 60% reductions) can better address air pollution exposure disparities, providing ranges of outcomes given modeling uncertainty. We consider exposure to individual racial/ethnic groups and racial/ethnic minorities overall (defined here as all except the non-Hispanic white population), using U.S. government statistics. We conclude by discussing policy implications, identifying where complementary policy approaches would be required to address equity-related air pollution concerns.

## Results

We explore whether selecting different sources of $CO_2$ reductions can better mitigate $PM_{2.5}$ disparities while achieving the same total $CO_2$ reductions, using optimization for constructing emissions reduction scenarios around a set of policy cases (see Table 1 for scenario names and scenario numbers). As a primary policy case, we estimate the distributional air quality impacts of a carbon policy in 2030 ("Cap 50%", scenario 3) relative to no carbon policy in 2030 ("Baseline", scenario 2) and the historical year 2017 ("Hist.", scenario 1). To ensure our primary policy case is a realistic projection, we draw from previous work that considered the potential impacts of a carbon pricing policy that reduces economy-wide emission in 2030 by 50% relative to the 2005 level. This scenario uses the outputs from an energy economic model of the U.S. economy combined with a power sector capacity expansion model (USREP-ReEDS) for a policy design that considers a range of technology cost assumptions and alternative emission allowance allocation schemes, detailed in the Methods section and Yuan et al.[24]. We then use a sensitivity scenario (scenario 4) to quantify the potential range for exposure reduction and equity outcomes for this particular policy due to uncertainty in the spatial distribution of sources, providing an upper and lower range for nationally averaged equity outcomes for each racial/ethnic group. We then conduct four different optimization scenarios (scenarios 5-8) around the primary policy case that test whether alternative emissions distributions that minimize racial/ethnic minority mortalities under different constraints can better mitigate $PM_{2.5}$ disparities while achieving the same 50% $CO_2$ reductions, and conduct two additional optimization scenarios (scenarios 9-10) testing the potential for mitigating disparities under 40% and 60% $CO_2$ reduction goals.

### Distributional air quality impact of carbon policy

Under the primary policy case, illustrating a carbon pricing policy, $CO_2$ emission reductions relative to *Baseline* in 2030 are driven mostly by

**Table 1 | Description of scenarios**

| Scenario number | Scenario name | Scenario category | Year | $CO_2$ targets | Method for specifying emissions from individual sources |
|---|---|---|---|---|---|
| 1 | Hist. (2017) | Primary | 2017 | NA | historical data |
| 2 | Baseline (2030) | Primary | 2030 | NA | scaling based on energy economic model output |
| 3 | Cap 50% (2030) | Primary (policy) | 2030 | −50% relative to 2005 | scaling based on energy economic model output |
| 4 | Uncertainty of Cap 50% (2030) | Sensitivity | 2030 | −50% relative to 2005 | constraints by ReEDS region-sector but varying point source reductions |
| 5 | Nation-sector | Optimization | 2030 | −50% relative to 2005 | minimizing racial/ethnic minority mortalities with constraint on total sectoral emissions reductions |
| 6 | State-total | Optimization | 2030 | −50% relative to 2005 | minimizing racial/ethnic minority mortalities with constraint on total state-level emissions reductions |
| 7 | State-sector | Optimization | 2030 | −50% relative to 2005 | minimizing racial/ethnic minority mortalities with constraint on sectoral emissions reductions for each state |
| 8 | Nation-total | Optimization | 2030 | −50% relative to 2005 | minimizing racial/ethnic minority mortalities with no constraints |
| 9 | Nation-total (−40%) | Optimization | 2030 | −40% relative to 2005 | minimizing racial/ethnic minority mortalities with no constraints |
| 10 | Nation-total (−60%) | Optimization | 2030 | −60% relative to 2005 | minimizing racial/ethnic minority mortalities with no constraints |

the electricity sector (77%), followed by transportation (10%), industry (7%), and residential and commercial sectors (6%). National emissions by sector in Hist., Baseline and Cap 50% (scenarios 1-3) are shown in Fig. 1. Changes vary regionally, with greatest absolute $CO_2$ reductions in Texas followed by the Alabama-Georgia-Tennessee region, and the largest reductions relative to *Baseline* in Idaho-Wyoming and West Virginia. In contrast, for states such as California and New York, ambitious state emission reduction targets are already in the *Baseline* and thus they experience few additional reductions under the federal policy. Regions and sectors with changes in $CO_2$ emissions also see changes in non-$CO_2$ emissions.

Changes in the electricity sector, with a near-elimination of coal-fired generation and additional reductions in other fuel combustion sources, drive reductions in sulfur dioxide ($SO_2$) and nitrogen oxides ($NO_x$), precursors to $PM_{2.5}$ formation in the atmosphere. Figure 1 shows that relative to *Baseline*, the policy reduces total emissions of $SO_2$ and NOx by 49% and 16%, respectively. For other pollutants where the electric sector is only a minor contributor to total emissions, reductions relative to the *Baseline* are smaller: 7% (primary $PM_{2.5}$), 1% (ammonia ($NH_3$)) and 5% (volatile organic compounds (VOC)). Emissions decrease under *Cap 50%* for each pollutant relative to Baseline. However, primary $PM_{2.5}$, $NH_3$, and VOC increase relative to their 2017 levels (Hist.).

Figure 2 shows simulated $PM_{2.5}$ concentrations (including primary and secondary $PM_{2.5}$) for Cap 50% (2030) (panel A), changes from Hist. and Baseline (panels B and C), and contributions by sector to changes from Baseline (panels D, E, F). $PM_{2.5}$ is simulated using the Intervention Model for Air Pollution (InMAP), with emissions inputs scaled following energy-economic model output. Relative to Baseline, the policy drives a reduction in total population-weighted average concentration by 0.37 µg/m³, with decreases in most, but not all counties and with changes ranging from −1.97 to +0.44 µg/m³. Reductions are greatest from Texas through the Mid-Atlantic region, driven largely by coal electricity emissions (panel D). Coal electricity emissions account for nearly half of the reduction in total average exposure (−0.16 µg/m³), with remaining reductions from transportation (−0.06 µg/m³), residential (−0.06 µg/m³), industrial (−0.05 µg/m³), non-coal electricity (−0.02 µg/m³), and food and agriculture (−0.01 µg/m³) (see Fig. S1 for other sectors). Although the Cap 50% scenario achieves reductions relative to Baseline, the average population-weighted concentration still increases relative to 2017 (see panel C) due to increases in activity levels.

Air pollution exposure decreases for all racial/ethnic groups in the primary policy case. Under Hist., average exposure for the total population was 7.2 µg m⁻³; racial/ethnic minorities experience higher exposure (8.0 µg m⁻³), and white populations slightly lower (6.7 µg m⁻³), shown in red in Fig. 3. Under Baseline, average exposures are slightly higher (7.8 µg m⁻³ for the entire population; 8.7 µg m⁻³ for racial/ethnic minorities overall). In 2030 under Cap 50%, average exposures are lower than Baseline for all racial/ethnic groups, with the greatest reductions for Black (0.44 µg/m³) and white populations (0.37 µg/m³).

Despite the overall reduction in $PM_{2.5}$ exposure in the primary policy case, it does not reduce exposure disparities at the national level. Relative exposure disparity (calculated as the percentage difference between the exposure for a given group and the total population) was 12.1% for racial/ethnic minorities and −6.9% for the white population under Hist., shown in blue in Fig. 3. Relative disparities increase for Asian, Hispanic, and racial/ethnic minorities, and disparities for Black and white populations decrease on average, relative to 2017. Reductions in exposure for the Black population and white population are greater than the reductions for the total population on average (0.37 µg/m³), thus reducing the relative disparity for Black population (from 17.9% to 17.8%) and increasing the average relative benefit for the white population (from −7.3% to

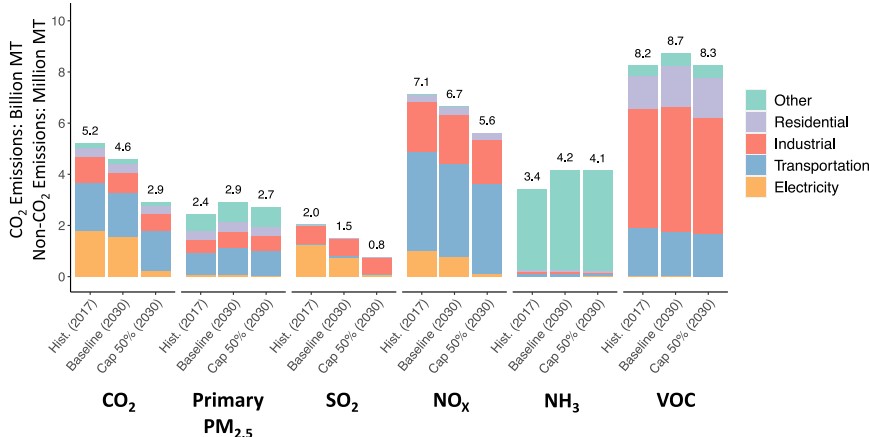

**Fig. 1 | National emissions under baseline and climate policy scenarios.** National emissions (billion metric tons (MT) for $CO_2$ and million MT for non-$CO_2$ pollutants) by pollutant and emission sectors in Hist. (2017), Baseline (2030), and Cap 50% (2030). Aggregated values are displayed above each bar.

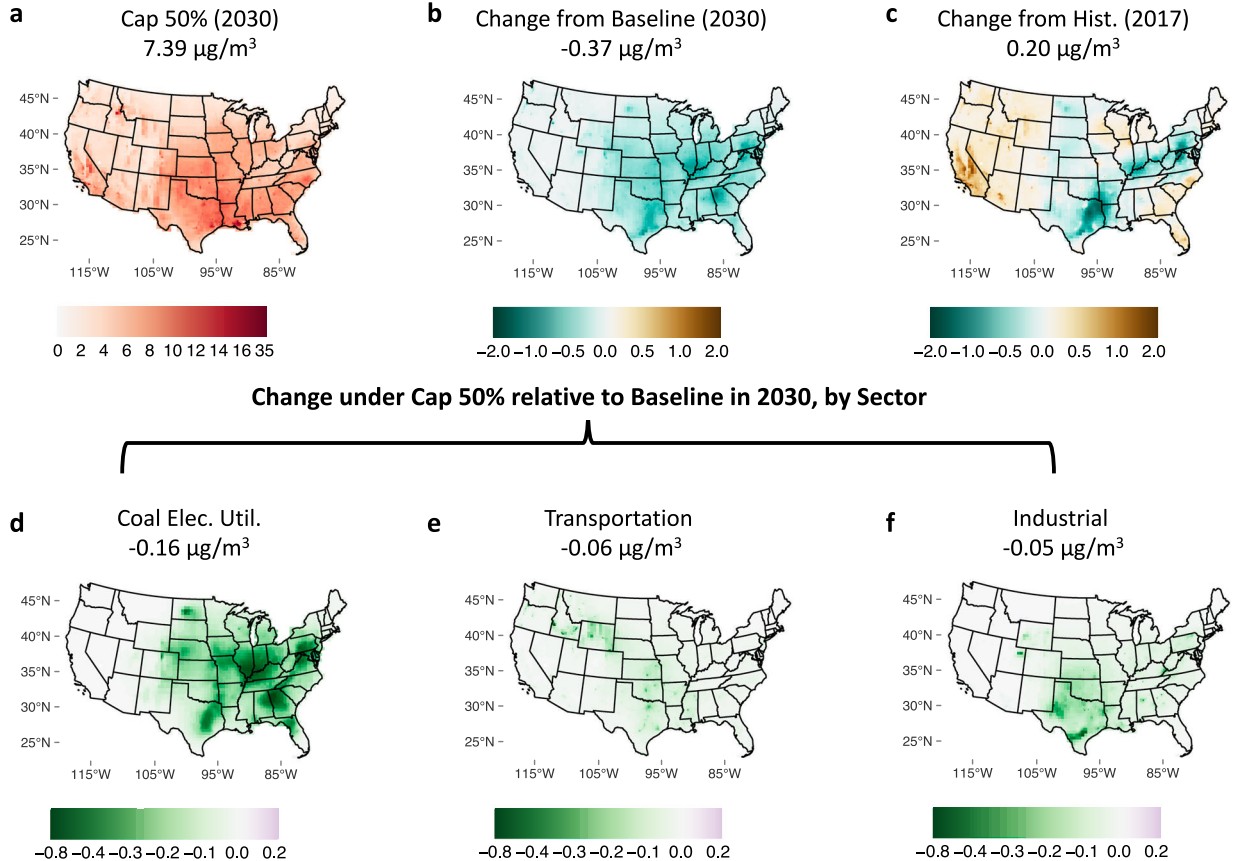

**Fig. 2 | Impacts of climate policy scenarios on PM$_{2.5}$ concentration over continental US. a–c** Annual average PM$_{2.5}$ concentrations (μg/m³) under Cap 50% (2030) and changes under Cap 50% (2030) relative to Baseline (2030) and Hist. (2017). **d–f** Change in concentrations under Cap 50% (2030) relative to Baseline (2030) from the three leading sectors. National population-weighted averages of (total or changes in) PM$_{2.5}$ concentration are listed under each respective title.

Changes from other sectors can be found in Figure S1. The base map of U.S. states is plotted using the R package tigris (https://CRAN.R-project.org/package=tigris), with original shape files from the U.S. Census Bureau (https://www.census.gov/geographies/mapping-files/time-series/geo/tiger-line-file.html, year 2019, resolution 20m).

−7.7%). In contrast, reductions in exposure for the Asian population (0.33 μg/m³), the Hispanic population (0.32 μg/m³), and for racial/ethnic minorities overall (0.36 μg/m³) are less than for the total population. As a result, the relative disparities increase for the Asian population (9.1% to 10.1%), the Hispanic population (12.1% to 13.3%) and for racial/ethnic minorities (12.4% to 13.1%), and the disparity gap between these groups and the white population widens slightly. Thus, while each group benefits on average from the carbon policy with lower average exposures, relative disparities mostly persist (or even increase).

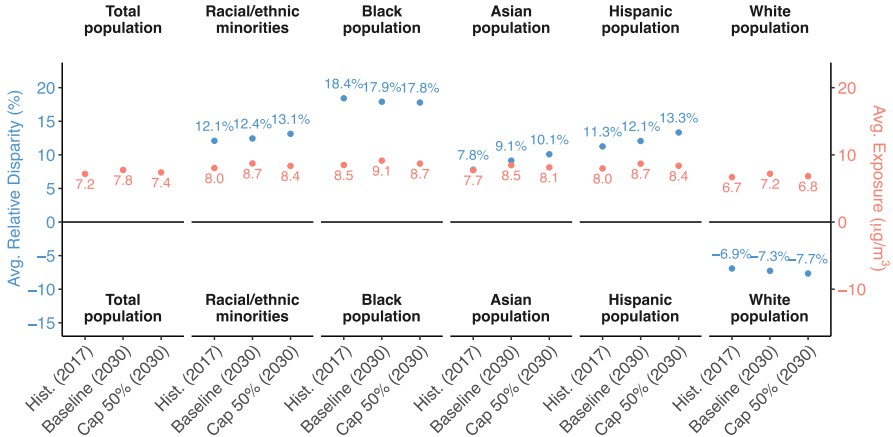

**Fig. 3 | Impacts of climate policy scenarios on population-weighted PM₂.₅ and relative disparities.** National population-weighted average PM₂.₅ exposure and relative disparity by race/ethnicity in Hist. (2017), Baseline (2030) and Cap 50% (2030). Disparity is calculated as the percentage difference between PM₂.₅ exposure for the given group and the total population. Racial/ethnic categories are derived from the American Community Survey.

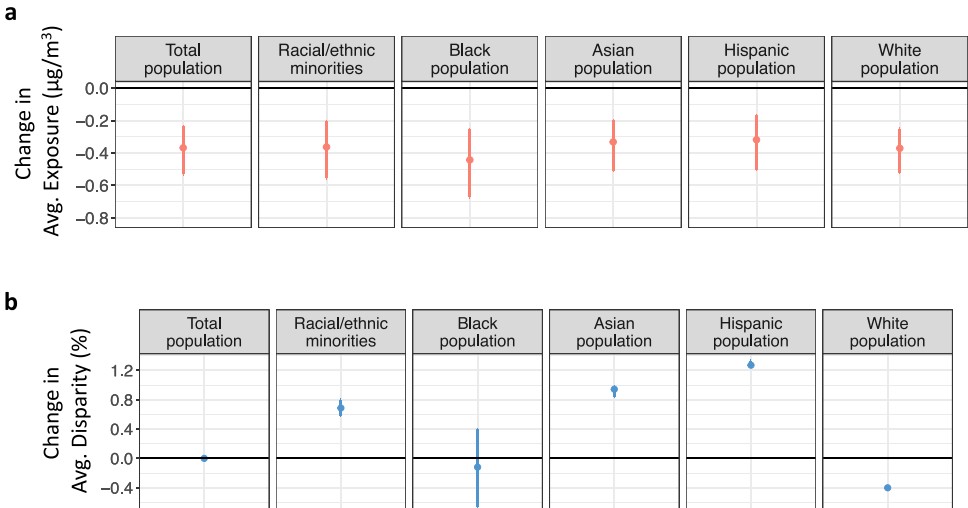

**Fig. 4 | Uncertainty in the estimated impacts on PM₂.₅ exposure and disparities.** Uncertainty range for the change in PM₂.₅ exposures (red, panel **a**) and disparities (blue, panel **b**) by race/ethnicity between Cap 50% (2030) and Baseline (2030). Results are based on two additional sensitivity simulations in which total reductions remain constant for each economic region and sector, but the distribution of these reductions among different point sources are allowed to vary. The error bars show the full range of changes in exposure and disparities across the three scenarios (main + two sensitivity scenarios). Disparity is calculated as the percentage difference between PM₂.₅ exposure for the given group and the total population. Racial/ethnic categories are derived from the American Community Survey.

As air pollution exposure disparities arise at multiple scales – national, regional, and local, we further explore the impacts of the primary carbon policy on exposure disparities within each state and within each urban area. We find a considerable amount of heterogeneity at the state and urban area level (see Figs. S2, S3). Figure S2 shows the change in disparities by state between Cap 50% and Baseline, showing large regional variation in impacts, driven by the correspondence between the population of each group and the location of largest reductions (as shown in Fig. S1). While the policy narrows disparities in some states, widening disparities in other states mean that there is limited aggregate impact at national scale. At urban scale, we focus on the 20 most populous urban areas in the US. We find that the carbon policy exacerbates the within-city pollution disparities by a small margin, with large heterogeneity across different urban areas (Fig. S3). We also find that the aggregated impacts of policy on exposure disparities are largely driven by its effects on the exposure disparities at the regional level (instead of at the local scale), consistent with findings from prior studies[25,26]. For example, while we find that the

policies exacerbate within-city exposure disparities for the Black population in almost all 20 major urban areas, the policy reduces the exposure disparities when aggregated; this is because regions with higher percentages of Black population generally experience a larger reduction (despite the smaller reduction within each urban area compared to other groups).

Our primary policy case assumes that the present-day emissions distribution for each sector within each of the regions simulated by the prior economic model analysis (see Methods) remains unchanged under Baseline and Cap 50%. However, emissions under carbon policies could change heterogeneously in ways that affect distributional outcomes. We assess with an additional scenario the degree to which our results change under this uncertainty for point sources (Table 1, scenario 4, see Methods/Uncertainty Analysis) to provide an upper and lower range for equity outcomes for each racial/ethnic group. Figure 4 shows resulting uncertainty ranges of changes in exposure and disparities by group between Cap 50% and Baseline. For all groups except the Black population, the impact of this change in distribution is

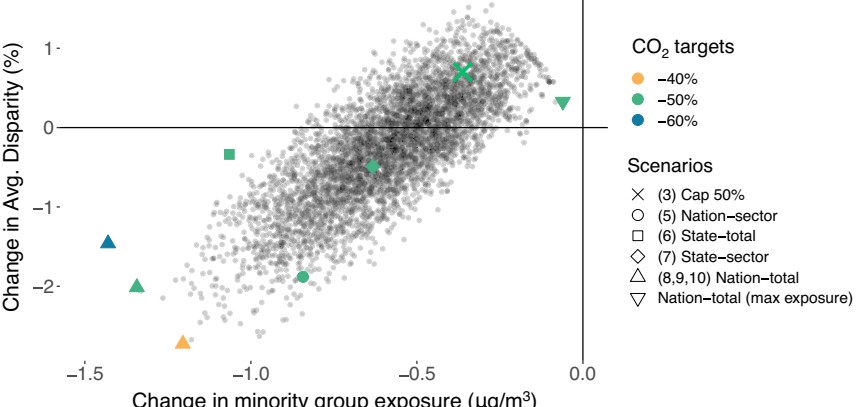

**Fig. 5 | Impacts of alternative scenarios on racial/ethnic minority exposure and disparities between racial/ethnic minorities and the overall population.** All results are relative to the 2030 baseline scenario (Table 1, #2). The green X shows the results of our main policy case (Cap 50%, Table 1, #3). The colored dots show the optimization scenarios (with associated numbers from Table 1) which minimize the racial/ethnic minority exposure under different constraints while achieving emissions reductions from 40% to 60% (emission targets are shown by colors). "Nation-total (max exposure)" shows a scenario that maximizes the racial/ethnic minority mortality while achieving 50% CO$_2$ reductions, serving as a lower bound of the exposure reductions. The black dots show 5000 potential emission reduction scenarios, derived from random selection of sources and the amount of emissions reductions at each source, which all achieve the same level of CO$_2$ reduction (−50%). Those scenarios do not aim to optimize the racial/ethnic minority mortalities but are used to show the full potential range of alternative policy scenarios.

relatively limited. For the Black population, the disparity can either increase or decrease depending on emissions distribution, although the magnitudes of relative changes remain small (0.5% relative to 18.4%).

## Potential for disparity mitigation through alternative carbon reduction distribution

The primary policy case described above illustrates the impact on disparities from regions and sectors that minimize CO$_2$ reduction cost (and associated spatial uncertainty). Next, we consider whether reducing the same amount of CO$_2$ from other combinations of regions and sectors could better mitigate air pollution exposure disparities, using an optimization approach. We conduct four different scenarios (Table 1, scenarios 5-8) to approximate different reduction strategies that might be achieved using either command-and-control or pricing mechanisms (see Methods/Optimization Approach). To do this, we use optimizations in which CO$_2$ reductions can come from different combinations of sources, minimizing PM$_{2.5}$ associated mortality for racial/ethnic minorities, with a variety of different constraints. In all optimizations, we minimize racial/ethnic minority mortality while keeping target CO$_2$ emissions reduction totals consistent under the different constraints and allowing individual sources to reduce in differing amounts to meet the overall target. Under "State-sector", overall state and sectoral reductions are the same as in the "Cap 50%" policy, but the distribution of reductions among individual sources within each sector-state combination can vary. "State-total" maintains consistent reductions for each state, but allows reductions to come from different economic sectors. "Nation-sector" maintains Cap 50%'s distribution of sectoral reductions but allows reductions from those sectors to come from anywhere in the country. "Nation-total" sets a U.S.-wide cap and allows any source to reduce to meet it. The "State-sector" and "State-total" scenarios could correspond to efforts that states might introduce to prioritize CO$_2$ reductions in specific locations based on knowledge of sources that contribute the most to racial/ethnic minority exposures. The least-constrained "Nation-total" scenario reflects a conceptual upper limit of the potential for targeting individual sources through national-scale policy design under a carbon reduction scenario of comparable magnitude. To test whether strengthening or weakening carbon reduction goals results in different outcomes, we also conduct "nation-total" scenarios for 40% and 60% reduction targets.

In Fig. 5, we compare these optimization scenarios to the impacts estimated for the primary policy case, Cap 50%, plotting the change in average disparity between the total population and racial/ethnic minorities vs. the change in racial/ethnic minority exposure. We show the different optimization constraints, as well as 5000 non-optimal scenarios for the 50% case which illustrates the range of potential outcomes from different source reduction choices. The 5000 non-optimal scenarios are derived from sampling a selection of sources and the amount of emissions reductions at each source in order to meet the same −50% CO$_2$ targets, without aiming to minimize racial/ethnic minority mortalities. To illustrate the potential range of results, we also show a scenario that maximizes racial/ethnic minority mortalities. We also show results from optimizations with national total 40% and 60% reductions. Numerical results for these scenarios are presented in Table S2.

While the primary policy scenario results in a widening of the air pollution exposure disparity for racial/ethnic minorities, further reductions in racial/ethnic minority exposures are in principle achievable while still meeting the same CO$_2$ emissions reductions. The comparison between the Cap 50% scenario and the additional scenarios in which reductions can come from alternate sectors and regions implies that the least cost reduction opportunities identified by the carbon policy do not produce the greatest improvements in PM$_{2.5}$ exposure. Prioritizing reductions in exposure for racial/ethnic minorities also reduces exposure for the white population and the total population on average, suggesting a win-win of absolute gains from reducing sources that minimize racial/ethnic minority mortality. However, this also means that the reduction in the overall disparity is limited, and substantial disparities remain. Allowing reductions to come from any source within a state can reduce disparities by 0.34% (the "State-total" scenario), while an additional 1.67% can be achieved by allowing reductions to come from different states (the "Nation-total" scenario). The sectoral contributions to this distribution are illustrated in Fig. S4; the largest driver of additional reductions comes from the optimization constraint that allows for redistribution of emissions in the transportation sector, which is not substantially affected under the "Cap 50%" policy but is largely responsible for the range of exposure reductions under the same CO$_2$ emission targets. Increasing the stringency of the

carbon policy to 60% further reduces racial/ethnic minority exposure, but also reduces exposure to the overall population, resulting in a smaller change in disparity relative to the baseline. A less ambitious (40%) carbon reduction, in contrast, results in a greater change in disparity, but less overall $PM_{2.5}$ exposure benefit.

## Discussion

We explored how federal decarbonization strategies might affect disparities in $PM_{2.5}$ exposure for different U.S. racial/ethnic groups, focusing on $CO_2$ policy of similar magnitude to current federal targets. We showed that a cap-and-trade policy instrument reduces exposure to $PM_{2.5}$ for all racial/ethnic groups relative to Baseline, but does not substantially mitigate relative disparities in exposure. Black, Hispanic, and Asian populations continue to experience disparities, while the white population experiences less exposure than the total population on average. This is because the carbon policy achieves most reductions in the coal-fired electricity sector. Previous studies have showed that this sector disproportionately harms only Black and white populations more than average[1]. In contrast, the electricity sector contributes a relatively small fraction to population exposure overall, and key disparities arise from sectors with remaining emissions even under 50% cuts, such as industry and heavy-duty diesel transportation. These results are robust to assumptions about emissions reduction distribution, suggesting that the geographic distribution of source reductions under comparable policies do not drive substantial differences in outcomes with respect to disparities. As shown in our optimization results (Fig. 5), the simulated carbon pricing scenario (primary policy case) achieves relatively smaller impacts on closing the disparities between racial/ethnic minorities and the total population, among the potential range of the alternative scenarios.

More broadly, we find limited opportunities to further reduce exposure and mitigate disparities at national scale while achieving the same $CO_2$ reduction goals. Even the "best" scenario for reducing pollution exposure disparities (among all the potential alternative scenarios) can only reduce the exposure disparities by a small portion (it reduces the disparity gap by 2.7 percentage points, compared to a disparity gap of 12%). Our analysis thus suggests that any scenario that aims to reduce $CO_2$ emissions by a similar magnitude is unlikely to substantially reduce existing $PM_{2.5}$ exposure disparities. The extent of air pollution mitigation is limited in part due to the magnitude of the $CO_2$ reductions desired by 2030, where addressing only 50% of $CO_2$ emissions leaves many polluting sources unmitigated. At the same time, efforts to prioritize reductions for racial/ethnic minorities benefit the entire population, including the white population, on average. We conclude that while reducing $CO_2$ by 50% can yield air pollution and health benefits for all, and has the potential to provide targeted improvements in particular regions, climate policy alone is an insufficient tool to adequately address near-term air pollution disparities nationally.

Our analysis contributes to a broader and emerging literature that documents the complex interactions between climate policies, overall air quality benefits, and pollution disparities. Our research reveals two insights about this complex interaction: first, we find that different decarbonization pathways can have different impacts on existing pollution disparities, both in terms of sign and magnitude. Second, we find that maximizing overall air quality benefits (even for disadvantaged communities) does not always help reduce the pollution disparities between different populations. Both insights are broadly consistent with previous papers that focus on different geographical regions, sectors, or policy contexts. For example, Mayfield studied the pathways to phase out coal power plants and found that the phase-out pathway that minimizes the cumulative mortalities from the electricity sectors is often not the highest-ranked pathway in terms of its impacts in reducing the pollution inequities measured using a suite of air quality equity indices the paper considered[27]. Goforth and Nock found

that without strict renewable energy and low carbon targets some electricity sector decarbonization pathways can exacerbate the pollution disparities[28]. These insights also hold beyond the electricity sectors[29]. For example, Zhu et al. evaluated two decarbonization strategies (building electrification and truck electrification) in California, and found that while building electrification generates greater overall air quality benefits (-15%), it is comparatively less beneficial to disadvantaged communities[19]. Polonik et al. find potential for targeted emissions reductions to reduce disparities, based on year 2017 emissions inventories and economic activity, especially for the transportation sector; our results, based on year 2030 projections, show more limited reduction possibilities[22]. Building on this previous work, our optimization results further demonstrate that one likely reason for these insights is the tension between applying large-scale policies to air pollutants, and reducing pollution disparities for specific disadvantaged communities which are affected by different sources (as also suggested by Wang et al.[30]).

This analysis considers reductions from sectors that are addressed in the IRA, which is expected to achieve U.S. carbon reductions through incentives targeted to clean energy and transportation[23]. With an incentive-based approach, $CO_2$ reductions from these sectors will not be specifically targeted towards addressing individual sources. Because we consider a comprehensive range of possible distributions of $CO_2$ reductions, our results are applicable to a variety of the reductions that might occur when the IRA is implemented. Analysis of the provisions of the IRA would be needed to specifically project its anticipated impact on air pollution and equity for different regions. However, as the entire range of potential $CO_2$ reduction distributions we assessed reduced air pollution exposure overall and also had limited impact on disparities at national scale, we would expect a similar outcome for the IRA.

In summary, we show that simply reducing $CO_2$ sources over the next decade with a magnitude comparable to the current U.S. federal policy target over the next decade, even if those sources are carefully selected, will not result in major reductions in air pollution exposure disparities among racial/ethnic groups in the U.S. Our results suggest several ways forward for policy design. Even with increased stringency, the emissions impacts of reducing $CO_2$ alone will not substantially change existing pollution disparities. This means that fulfilling policy goals associated with minimizing disproportionate impacts of air pollution on different racial/ethnic groups will require additional targeted interventions in the near term. More aggressive carbon policies than examined here, including those that ultimately remove all fossil fuel sources, could have larger effects, but the timescale of this transition would leave disparities unaddressed for more than a decade. Interventions to reduce both direct $PM_{2.5}$ and precursor emissions that are not directly associated with $CO_2$ sources, such as sectoral policies and community-focused mitigation measures, will be critical to improving air quality and public health equitably in the U.S. Taken together, this suggests that efforts fully mitigate the disparate impacts of pollutants will require efforts beyond optimization of existing $CO_2$ policy strategies, including large-scale structural changes.

## Methods

In this section, we first describe the energy-economic modeling of the baseline and carbon pricing scenarios that project the energy sector activity in 2030. We then estimate future levels of emissions, using the projected energy sector activity to scale historical U.S. emissions of $CO_2$, primary $PM_{2.5}$, and precursor gases that form secondary $PM_{2.5}$ in the atmosphere – sulfur dioxide ($SO_2$), nitrogen oxides ($NO_X$), ammonia ($NH_3$), and volatile organic compounds (VOC). Non-$CO_2$ emission factors per unit of energy used are fixed at 2017 levels to enable consistent comparisons, as we do not have information regarding how non-$CO_2$ emission rates per unit of energy used will change over time. Using these emissions, we then apply a reduced-form air quality model to estimate annual $PM_{2.5}$ concentrations and

population exposures at a fine spatial scale and evaluate relative exposure disparities across racial/ethnic groups. Finally, we evaluate a large set of potential emission reduction scenarios that achieve the same level of $CO_2$ reductions, and thus calculated the full impacts on total pollution exposure and the pollution disparities between different population groups.

### Primary policy case

The primary policy case is constructed from two future scenarios for 2030, described in detail by Yuan et al.[24]: a national $CO_2$ cap-and-trade program that requires a 50% reduction in U.S. economy-wide $CO_2$ emissions relative to 2005 levels by 2030 ("Cap 50%", scenario 3 in Table 1), and (2) a baseline scenario without the program ("Baseline", scenario 2 in Table 1). Yuan et al. deploy an economy-wide, energy-economic modeling tool (USREP-ReEDS) to evaluate the impact of potential $CO_2$ pricing policies on energy sector activity, $CO_2$ emissions, household welfare, and total net benefits. MIT's U.S. Regional Energy Policy (USREP) model is a computable general equilibrium model of the U.S. economy[10], and in these simulations its electricity sector representation has been replaced by the Regional Energy Deployment System (ReEDS), a capacity expansion model of the U.S. electricity sector developed by the U.S. National Renewable Energy Laboratory (NREL)[31]. Relevant to air pollution projections in this paper, USREP represents states via 30 regions (including 18 individual states), while ReEDS spans 134 electricity balancing regions (with additional geographic representation of wind and solar resources across 356 regions).

In the Baseline scenario ("Baseline"), results are calibrated to the Energy Information Administration's Annual Energy Outlook 2020 reference case and in addition, reflect NREL's Annual Technology Baseline 2019 Mid-Range electricity technology costs and performance characteristics, updated state clean energy policies, and a COVID-19 pandemic adjustment. The policy scenario ("Cap 50%") imposes on the Baseline a national $CO_2$ cap-and-trade program that covers energy and industry-related $CO_2$ emissions and allows national trading of emissions allowances (at a price of $14 in 2025 and rising to $99 in 2030) without offsets or banking or borrowing across years. The scenario assumes that $CO_2$ emission allowances are distributed to states on a per-capita basis and that the state revenue raised from allowance sales are rebated to households on a per-capita basis. While other choices of allowance allocation schemes are evaluated by Yuan et al. affected economic welfare outcomes[24], they have negligible impact on emissions outcomes and therefore are not analyzed here.

### Emissions inventory

We construct emissions inventories for a base historical year ("Hist.," 2017) and the modeled Baseline and Cap 50% scenarios in 2030, and take steps to make them compatible with the air quality model that we use (discussed in the following section).

We use the U.S. Environmental Protection Agency's (EPA) National Emission Inventory (NEI) 2017 containing annual emissions of $CO_2$, $PM_{2.5}$, $SO_x$, $NO_x$, $NH_3$, and VOC for 5,495 unique EPA Source Classification Codes[32]. We use emissions spanning the continental U.S., allocating emissions spatially to grid cells and vertically to effective stack height (ESH) layers (reflecting the height of the emission plume that rises above the physical stack height). For point sources, we use the unique coordinates of each point source to assign the corresponding grid cell that each source is located in. We calculate ESHs for each point source using stack information (height, diameter, plume velocity, and plume temperature) applying the Holland formula[33], using ambient temperature and wind speed from the air quality model's atmospheric layer that corresponds to the emission source's stack height and location, and ambient pressure that we calculate as a function of sea level temperature and real stack height. If a source's stack height data is missing, we use the ESH layer of the nearest source within the same NEI Tier 2 category.

For area sources, which are county-level and often overlap with multiple grid cells, we distribute emissions to grid cells using distributions in the NEI 2014 spatial modeling data prepared for use in Tessum et al.[8], as 2017 emission spatial distributions were not available. NEI 2014 distributions reflect spatial surrogates unique to specific emission types (e.g., population for dry cleaning emissions and interstate highways for motor vehicle emissions), that are used in development of EPA emissions modeling platforms. We distribute state-level NEI 2017 emissions to grid cells based on the state-grid distribution for the corresponding NEI Tier 3 emissions in the 2014 dataset. For cases where there is not a Tier 3 match, we use Tier 2 or Tier 1 distributions to allocate remaining 2017 emissions. We then assign all area sources the ground level ESH. Finally, biogenic and wildfire emissions are from 2005 and held constant[8]. The 2017 NEI includes $CO_2$ emissions for many point sources from the EPA's Greenhouse Gas Reporting Program (GHGRP) as well as for transportation area sources (calculated from EPA's MOVES model). While the GHGRP does not include all sources of emissions, it includes emissions from large facilities and in total covers approximately 85-90% of all U.S. GHG emissions[34]. We retain the $CO_2$ emissions for use in our sensitivity scenarios and optimization described below.

### Emissions projections

For the two future scenarios, we scale 2017 emissions to 2030 based on projected outcomes modeled with USREP-ReEDS, assuming that non-$CO_2$ emission factors (i.e. emission per unit output) are fixed at 2017 levels. The scaling approach largely follows methods outlined by[10]. All emissions – except power sector $CO_2$, $SO_2$, and $NO_x$ pollutants from coal and gas fuel sources – are scaled within 29 USREP regions (Alaska is excluded) and using 20 USREP variables matched to NEI Source Classification Codes, producing 545 unique scaling combinations nationally (35 region-variable combinations have zero data). Then, the scaling factor is applied uniformly to emissions of each pollutant (including $CO_2$) within the region and emissions scaling category. For the electricity sector, we scale coal and gas power plant emissions for $CO_2$, $SO_2$ and $NO_x$ to match ReEDS emissions for 134 balancing areas. Furthermore, total $CO_2$ emissions are then adjusted by USREP region by broader sectors (electricity, transportation, industrial, and residential) to match $CO_2$ emissions output by USREP, reflecting modeled efficiency improvements over time. We also perform a sensitivity analysis to evaluate the effects of spatial uncertainty of emission reductions on the projected impacts on exposure disparities. For our base case, we apply a uniform scaling factor to all emission sources from a specific sector within each USREP-ReEDS region (which have locations specified by the NEI). To address the spatial uncertainty of estimated emissions reductions under the uncertainty scenario (Table 1, #4), we produce alternative emissions distributions that are consistent with $CO_2$ emissions reductions in the energy-economic modeling but allow point source emissions to vary within each USREP-ReEDS region for each sector. Specifically, we optimize point source emissions changes under the carbon policy to estimate upper and lower bounds on mortality by race/ethnicity (see below for health impacts analysis and optimization methods), keeping total changes in $CO_2$ consistent by sector and region within the primary policy case. This redistribution of emissions is applied to the policy case only to evaluate a range of impacts due to the policy; the baseline case remains the same.

### $PM_{2.5}$ modeling, population exposure, and disparity metric

We estimate annual average concentrations of $PM_{2.5}$ for each scenario using the Intervention Model for Air Pollution (InMAP). InMAP is a reduced complexity air quality model (RCM) that reflects atmospheric chemistry and transport of particulate air pollution[35]. The model takes a set of emissions data (primary $PM_{2.5}$, $SO_x$, $NO_x$, $NH_3$, and VOC), among other inputs, and predicts annual average concentrations of total $PM_{2.5}$ and its components: primary $PM_{2.5}$, particulate sulfate

(pSO$_4$), particulate nitrate (pNO$_3$), particulate ammonium (pNH$_4$), and secondary organic aerosols (SOA). InMAP provides relatively higher spatial granularity than other RCMs or CTMs, while reducing the temporal resolution to annual scale (among other simplifications) to avoid computational requirements from more complex CTMs. InMAP has been used and validated in numerous peer-reviewed analyses of air quality and equity impacts of emissions[8,36,37]. RCMs, including InMAP, have been evaluated against each other and more sophisticated CTMs[38]. The reduced-form air quality modeling approach is limited by its largely linear chemical mechanism and its use of annual-averaged meteorology. However, previous studies have shown that regional nonlinearities are limited in the US[39], and that large-scale conclusions from InMAP modeling are comparable to those using more detailed chemical transport modeling[17].

In our analysis, we specifically use the InMAP Source Receptor Matrix (ISRM) as provided by[36]. Given emissions inputs of primary PM$_{2.5}$, SO$_x$, NO$_x$, NH$_3$, and VOC, the ISRM provides the change in respective particulate concentrations in a "receptor" grid cell caused by a 1 unit increase in emissions of each pollutant in a "source" grid cell. The sum of particulate concentrations of primary PM$_{2.5}$, pSO$_4$, pNO$_3$, pNH$_4$, and SOA equals total PM$_{2.5}$ in each grid cell. The ISRM spatially consists of 52,411 grid cells with resolutions ranging from 1 x 1 km (in the most population-dense areas) to 48 x 48 km (in the least population-dense areas), and vertically distinguishes between three ESH layers: "ground" 0–57 m, "low" 57–379 m, and "high" > 379 m. Emissions inputs – allocated to ISRM grid cells and ESH layers - are multiplied by the respective pollutant source-receptor matrix to produce concentrations of final PM$_{2.5}$ in each of the grid cell.

The ISRM includes block-group level population data by race/ethnicity from the 5-Year 2012 American Community Survey (ACS) that have been allocated to grid cells. Following Tessum et al.[1], we evaluate outcomes for several racial/ethnic groups: Asian, Black, Hispanic, racial/ethnic minorities, and non-Hispanic white groups. Here, Hispanic spans all races; Asian, Black, and white groups are non-Hispanic and correspond only to the specific race; and racial/ethnic minorities is everyone except the non-Hispanic white population. The sum of racial/ethnic minorities and white populations therefore equals the total population. Using total population projections from University of Virginia[40], we scale population data to 2030 by applying state level growth rates for the total population to all populations in grid cells whose spatial centroids correspond to a given state. We calculate a relative disparity metric at the national and state levels as the percentage difference between the average exposure for each group and the average exposure for the total population. We also calculate percentage point differences between the policy and baseline scenarios to evaluate how disparities change due to the policy.

### Optimization approach to assess alternative carbon reduction distribution

To explore the impacts of alternative scenarios that achieve the same level of CO$_2$ reductions on pollution exposure and disparities, we design an optimization approach to explore if emissions distributions that are different than those under the modeled carbon policy better mitigate national-scale air quality disparities while still achieving the same total CO$_2$ emissions reductions. To do this, we apply the following optimization methodology to minimize racial/ethnic minority mortality while keeping CO$_2$ constant for respective emissions group combinations: "State-sector", "State-total", "National-sector", "National-total." The sectors considered here are electricity, transportation, industry, and residential sectors.

First, using the ISRM, we calculate marginal mortality values (total U.S. mortality caused per ton of emissions of primary PM$_{2.5}$, SO$_2$, NO$_x$, NH$_3$, and VOC) for emissions from each grid cell for racial/ethnic minorities, using the concentration response function from Krewski et al.[41] and all-cause mortality incidence rates for the total population.

By matching emissions to their respective marginal mortality values, we can then calculate the mortality caused by each source and pollutant.

We conduct the optimization method following the following equation. In the optimization approach, emissions that are eligible to vary are sources that (1) cause PM$_{2.5}$-related mortality; and (2) have non-zero CO$_2$ emissions in the 2030 baseline.

Maximize or minimize:

$$\text{objective function} = \sum_i S_i TM_i \tag{1}$$

Subject to: $\sum S_i CO2_i = CO_2$ target and $0 \leq S_i \leq 1$ where:
- $i$ denotes the unique index of eligible emissions sources.
- $TM_i$ denotes total mortality (for racial/ethnic minorities) caused by emissions at source i in the 2030 baseline.
- $S_i$ denotes the scaling factor (decision variable) applied to emissions of all pollutants at source i, allowed to range between 0 and 1. A source is thus allowed to be completely shut down emitting zero emissions (i.e. $S = 0$), or emit as much as the baseline emission (i.e. $S = 1$).
- $CO2_i$ is the amount of CO$_2$ emitted by source $_i$ in the baseline 2030 scenario.
- CO$_2$ target denotes the fixed total CO$_2$ targets (40 to 60% reductions relative to the 2005 level).

The two constraints require that 1) total CO$_2$ emission reductions are fixed at a constant level (40 to 60% reductions relative to the 2005 level); 2) emissions of any pollutant cannot be less than 0 (lower bound) and cannot be higher than the level in the 2030 baseline scenario (upper bound).

### Reporting summary

Further information on research design is available in the Nature Portfolio Reporting Summary linked to this article.

## Data availability

All data that support the findings of the paper is presented in the paper, supplementary information, and public data repository. Main results and input data used in this study are available at: https://zenodo.org/record/8226507.

## Code availability

Code scripts used in this study are available at: https://zenodo.org/record/8226507.

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

## Acknowledgements

This publication was made possible by USEPA grant RD-83587201 (N.E.S., J.R.). Its contents are solely the responsibility of the grantee and do not necessarily represent the official views of the USEPA. Further, USEPA does not endorse the purchase of any commercial products or services mentioned in the publication. We gratefully acknowledge additional financial support for this work provided by the Hopewell Fund (J.R.). M.Q. was also supported by the planetary health fellowship from Stanford University.

## Author contributions

N.E.S. designed the research. P.P. performed air quality modelling simulations and calculated the impacts on pollution disparities. M.Q. and P.P. performed the optimization analysis. P.P., M.Q., N.E.S., S.D.E., M.Y., and J.R. interpreted the results and wrote the paper.

## Competing interests

The authors declare no competing interests.
