## [Peer Review File · Nature Communications]

Air Quality-Related Equity Implications of U.S. Decarbonization PolicyREVIEWER COMMENTS

Reviewer #1 (Remarks to the Author):

This paper investigates the potential air pollution exposure reductions that would result from U.S. federal carbon policy, and evaluates the disparities across racial groups. I think the topic is interesting and timely. Below my comments are aimed at helping the authors improve the paper.

In the abstract the authors say the gap widens, but no specifics are given. Please add in specific numbers this will make the abstract more understandable. Also this sentence is so general that I find it confusing "Alternative choices of sources that reduce a similar amount of CO2 emissions also cannot substantially mitigate these disparities." What are "alternative choices of sources"?

In the introduction there should be more discussion of environmental justice given that it is in the title of this paper. The following sentence in particular is so high level that it leaves the reader wondering what EJ goals the authors are talking about. "Addressing disparities in air pollution exposure and climate change risks are both closely tied to existing environmental justice (EJ) related policy goals." For example are you focusing on distributional or recognitional environmental justice? I think distributional. Also it is mentioned that Biden and Harris have EJ related goals, but those goals are not explicitly stated. Stating the specific goals would help this paper remain relevant even after there is a change in office.

It is unclear if the authors are arguing that a CO2 policy is as good as explicit EJ policies. If the authors are stating this I would have to strongly disagree because I think actors will try to game the system, and not all decarbonization policies will reduce local emissions in the same way.

This sentence intro was confusing: "We focus on reductions in economy-wide emissions by 50% below 2005 levels by 2030." Maybe reframe to spread out the numbers. For example: "We focus on reductions of at least 50% in 2030 economy-wide emissions, when compared to 2005 emission levels."
"

I think in the results there need to be a few more sentences on how your "policy design follows an energy-economic analysis conducted and described by Yuan et al. (2022)." Each paper should stand alone, so I think it would be great to add 2-3 sentences on the type of model used and what it does. This other sentence is also problematic because I the reader had to go read the other paper to understand this one "The inputs and results of the underlying energy-economic model scenario were described previously (Yuan et al., 2022)." Please add in text describing the scenarios.

In the results I think there needs to be a table, because it has a lot of moving parts. I found "Potential for Disparity Mitigation through Alternative Carbon Reduction Distribution" section to be a bit confusing. There are so many options, and none of them are explained in detail.

Also I feel like this paper has so few scenarios. It seems that the bulk of the work was done in the first paper (Yuan et al 2022) so I am wondering if the only add on here was the racial group analysis. There were many talked about in the political landscape. What about the renewable portfolio scenarios, or the low carbon standards? The 100% Renewable energy by 2035 or 2050.

Why are some regions analyzed at the census tract level and others at the county level? It was unclear which is which.

Fig 1 What type of electricity generation are you projecting because the PM looks very small but the SOx is very high. I think in the results before you show figures there needs to be some discussion of how the industries are changing in each of your scenarios. Right now there is no context for how to interpret these numbers. For a stylistic comment, the words at top were hard to read and connect at

first (same stylistic comment for fig 3). I suggest moving them to the bottom of the x axis so the reader can more easily see the emissions you are referring to.

Figure 2 in the main looks the same for panels e, h, and i. I think if this wants to stay on the main it should have different color bar or be a different figure. I think it doesn't make sense to send people to the SI. Also for this to be in the main the authors should discuss the results from these ones. What can we infer from these graphs?

Figure 4. I think this graphic does not convey your message well. If this is printed in black and white people will not be able to see the difference. I suggest adding in symbols. Also the title for racial groups should be moved to the X axis. It is hard to see the differences because there is overlap and the three symbols are all the same. I suggest changing to error bars or giving the different policies different symbols.

Why is your methods named "Online methods"?

An equity outcome is not defined. I think this paper needs to state what and equity outcome for the racial/ethnic groups means in plain English. Also I think this paper relies so heavily on the two previously published papers that it makes this one hard to read. For example in the carbon pricing policy the CO2 pricing policy (i.e., what level is the price), was not mentioned. I suggest a table with the relevant information about your scenarios, and potentially adding more scenarios.

Need more information on how you downscale from ReEDS regions which are very large to the census tract level.

I am not sure why in the methods the S scaling factor changes for each pollutant. Also in the methods the authors appear to contradict themselves. "while in the uniform scaling method S_i is uniform across all emission sources within a region and scaling variable set, here S_i is unique to each emissions source i as determined by the optimization." I suggest reframing this sentence. Also if you have change this to have S not be uniform then this is no longer the uniform scaling method.

The equation should be given in "Optimization Approach to Assess Alternative Carbon Reduction Distribution" for the objective function and constraints.

I think there should be more discussion of the different sectors and their impacts on vulnerable communities in the authors results section. Which sector has the biggest impact on disparities.

I would like to bring this recent paper in Nature communications, and another paper which appears similar, but a big difference is these paper below focus solely on the electricity sector. I think the authors should compare their results.

- Goforth, T., Nock, D. Air pollution disparities and equality assessments of US national decarbonization strategies. Nat Commun 13, 7488 (2022). <https://doi.org/10.1038/s41467-022-35098-4>
- Mayfield, E. N. Phasing out coal power plants based on cumulative air pollution impact and equity objectives in net zero energy system transitions. Environ. Res. Infrastruct. Sustain. 2, 021004 (2022). <https://iopscience.iop.org/article/10.1088/2634-4505/ac70f6>

Reviewer #3 (Remarks to the Author):

This is an outstanding paper with important conclusions. The methods are state-of-the art and appropriate for the questions at hand. The results are of compelling importance for reaching the stated aims of the US Government to reduce air pollution disparities as a cornerstone of its climate

policy.

I recommend publication with the addition of some minor additional analyses and discussion.

- A comment. I found one aspect of Figure 1 quite remarkable -- the reductions in CO₂ from this climate policy greatly outpace the reductions in the emissions of all of the other pollutants. And it's not so surprising -- it's well known that non-energy-related emissions play a major role in present-day exposures (see for example the flow diagrams of Thakrar et al ES&T Letters 2020). Among those pollutants that do have emissions reductions, it's also interesting to note that only NO_x and SO₂ really meaningfully reduce (and less than CO₂). I can see why this might be the case for a suite of policies highly focused on transport and electric power generation. So, my suggestion:

You might speak to the degree to which the results of your analysis are strongly driven by the emissions aspects of the story. Do we see limited disparity benefits from this simulated policy largely because it is considerably more effective at reducing CO₂ than emissions of PM precursors?

- Suggested supplementary analysis on spatial scales. Considerable recent work shows that PM_{2.5} disparities arise from multiple different spatial scales -- regional differences in demographics (e.g., the Southeastern US has higher PM and higher share of Black population); urban-rural differences (urban areas have higher PM and are more diverse) and finally the within-urban differences in exposure that arise due to forces of urban segregation and land use policy. All three spatial scales matter, of course, in contributing to inequalities. However, it is arguably the within-urban inequalities that often attract the largest attention, perhaps simply because of how readily apparent and ethically troubling they are. So, my suggestion:

You might consider a minor supplemental analysis to look at the degree to which your results differ when considered at different spatial scales. For example, you could do a within-urban sub-analysis by considering the distribution of disparity reduction in urban areas only by clipping to the InMAP grid cells that correspond to US census urbanized areas, and comparing the distribution of changes in each city. I don't think this needs more than an SI figure or two.

PS - this comment was inspired by reading a new paper just published in ES&T Letters: Liu J and Marshall JD, DOI: 10.1021/acs.estlett.2c00826

- Suggested addition to your discussion/conclusions: I find that this paper adds to a growing body of evidence that systemic racial-ethnic inequalities in air pollution exposure are unlikely to be resolved simply as a consequence of air pollutant emissions reductions. One relevant recent paper on this topic you might be interested in is Wang et al, PNAS 2022 (10.1073/pnas.2205548119). I think a reasonable working hypothesis for this broadly consistent result is that emissions sources are disparately concentrated near communities of color, and thus an economy-wide reduction of emissions does not eliminate relative disparities. So, my suggestion:

I think your conclusion section could state even more strongly your key result. In plain terms, expecting that climate policy will naturally lead to major reductions in relative racial/ethnic PM_{2.5} disparities runs contrary to the available evidence, and your valuable simulation provides a compelling demonstration of this point.

- Comment: I think your research very nicely tees up the questions of (i) why do these results arise, and (ii) how might we better design decarbonization policies to reduce disparities, or (iii) perhaps these really need to be pursued in parallel with distinct approaches? These questions are very timely for current activities in US EPA and in the Biden Administration.

General Response:

We appreciate the opportunity to revise our manuscript in response to these detailed reviews. In summary, we believe that the substantial changes we made have strengthened the manuscript, and provide additional clarity and support for our methods and conclusions. Specifically, we have added additional scenarios with appropriate details, included additional discussion of our results in the context of environmental justice, and reframed the conclusions to center on the impact and policy conclusions. On additional scenarios, we have also clarified how this work relates to previously published work: our main results rely on an optimization method, which is substantively different from the economic model driven analysis reported previously which we use as a starting point and compare our results to. We hope this makes the distinction more obvious.

We have also thoroughly reviewed the manuscript and SI and made edits to fix typos, references, and captions throughout the text.

Below, we include a detailed response to the reviewer's comments. Reviewer comments are in plain text, our responses are in italics, and changes to the manuscript are in blue.

REVIEWER COMMENTS

Reviewer #1:

This paper investigates the potential air pollution exposure reductions that would result from U.S. federal carbon policy, and evaluates the disparities across racial groups. I think the topic is interesting and timely. Below my comments are aimed at helping the authors improve the paper.

In the abstract the authors say the gap widens, but no specifics are given. Please add in specific numbers this will make the abstract more understandable. Also this sentence is so general that I find it confusing “Alternative choices of sources that reduce a similar amount of CO₂ emissions also cannot substantially mitigate these disparities.” What are “alternative choices of sources”?

Response: *We have changed language in the abstract to make it more clear that we use an optimization approach, which assesses every possible combination of CO₂ reductions within specified constraints, and we calculate the resulting implications for disparities. We have also added specific numbers to the abstract for our primary policy case, reducing other text to ensure the abstract remains within the 150-word limit.*

The relevant new text in the abstract reads:

The 50% CO₂ reduction policy case reduces average fine particulate matter (PM_{2.5}) exposure across racial/ethnic groups, with greatest benefit for non-Hispanic Black (-0.44 µg/m³) and white populations (-0.37 µg/m³). The average exposure disparity for people of color rises from 12.1% to 13.1%. Applying an optimization approach to multiple emissions reduction scenarios, we find that no alternate combination of reductions from different CO₂ sources would substantially mitigate exposure disparities. Results suggest that CO₂-based strategies for this range of reductions are insufficient for fully mitigating PM_{2.5} exposure disparities between white and non-white populations; addressing disparities may require larger-scale structural changes.

In the introduction there should be more discussion of environmental justice given that it is in the title of this paper. The following sentence in particular is so high level that it leaves the reader wondering what EJ goals the authors are talking about. “Addressing disparities in air pollution exposure and climate change risks are both closely tied to existing environmental justice (EJ) related policy goals.” For example are you focusing on distributional or recognitional environmental justice? I think distributional. Also it is mentioned that Biden and Harris have EJ related goals, but those goals are not explicitly stated. Stating the specific goals would help this paper remain relevant even after there is a change in office.

Response:

We did not include reference to environmental justice in the title of the paper – the title remains as in the previous submission "Air Quality-Related Equity Implications of U.S. Decarbonization Policy" as we specifically examine air quality exposure disparities. We agree with the reviewer that the broad link to environmental justice is complex, and we do not focus explicitly on making that link. Thus, we have removed reference to EJ as a

concept in the sentence that the reviewer highlights and in the key words, instead being specific about the exposure inequities we are assessing in the paper.

The goals announced by the US government in January 2021 are now explicitly stated. We have altered language in the introduction to emphasize existing US national policies and commitments, rather than referencing the current administration specifically.

The new language is the following:

Addressing disparities in air pollution exposure to people of color and low-income populations and mitigating climate change risks are both closely tied to existing policy goals: In January 2021, the US government announced a target that 40% of the overall benefits of certain federal investments, including investments in the areas of clean energy, should flow to disadvantaged communities ⁴. Here, we use energy-economic scenarios and an air quality model to quantify whether and how different policies that reduce carbon dioxide (CO₂) emissions by 40-60% in 2030 relative to 2005 levels simultaneously reduce racial and ethnic air pollution disparities at national scale. A 50% reduction by 2030 is consistent with the U.S. pledge under the Paris Agreement (which aims to reduce emissions by 50-52% compared to the 2005 level) and the 2022 Inflation Reduction Act (IRA) ⁵. We evaluate the extent to which carbon policies of comparable magnitude and sectoral scope can feasibly achieve reductions in air pollution disparities.

It is unclear if the authors are arguing that a CO₂ policy is as good as explicit EJ policies. If the authors are stating this I would have to strongly disagree because I think actors will try to game the system, and not all decarbonization policies will reduce local emissions in the same way.

Response: *We are, in fact, arguing the opposite – CO₂ policies will not address the issue of air pollution inequities – and we regret that we were not able to fully communicate this point in the previous draft. In the abstract, the final sentence has been modified to more clearly state this conclusion:*

Results suggest that CO₂-based strategies for this range of reductions are insufficient for fully mitigating PM_{2.5} exposure disparities between white and non-white populations; addressing disparities may require larger-scale structural changes.

We have also substantially edited our discussion to make this central statement more clearly. For example, in the conclusion paragraph (the last paragraph of the main text), we summarised the central policy implications of our paper:

In summary, we show that simply reducing CO₂ sources over the next decade with a magnitude comparable to the current U.S. federal policy target over the next decade, even if those sources are carefully selected, will not result in major reductions in air pollution exposure disparities among racial/ethnic groups in the U.S. Our results suggest

several ways forward for policy design. Even with increased stringency, the emissions impacts of reducing CO₂ alone will not substantially change existing pollution disparities. This means that fulfilling policy goals associated with minimizing disproportionate impacts of air pollution on different racial/ethnic groups will require additional targeted interventions in the near term. More aggressive carbon policies than examined here, including those that ultimately remove all fossil fuel sources, could have larger effects, but the timescale of this transition would leave disparities unaddressed for more than a decade.

This sentence intro was confusing: “We focus on reductions in economy-wide emissions by 50% below 2005 levels by 2030.” Maybe reframe to spread out the numbers. For example: “We focus on reductions of at least 50% in 2030 economy-wide emissions, when compared to 2005 emission levels.”

Response: *We thank the reviewer for the suggestion. We have changed the sentence, and it now reads:*

We focus on 40-60% reductions in economy-wide emissions by the year 2030, when compared with 2005 levels.

I think in the results there need to be a few more sentences on how your “policy design follows an energy-economic analysis conducted and described by Yuan et al. (2022).” Each paper should stand alone, so I think it would be great to add 2-3 sentences on the type of model used and what it does. This other sentence is also problematic because I the reader had to go read the other paper to understand this one “The inputs and results of the underlying energy-economic model scenario were described previously (Yuan et al., 2022).” Please add in text describing the scenarios.

Response: *We have now clarified how our results relate to the previous paper by Yuan et al., specifying that we draw our primary policy case from their results. We have also added detailed explanations of the modeling scenarios in the method section and Table 1.*

We explore whether selecting different sources of CO₂ reductions can better mitigate PM_{2.5} disparities while achieving the same total CO₂ reductions, using optimization for constructing emissions reduction scenarios around a set of policy cases (see Table 1 for scenario names and scenario numbers). As a primary policy case, we estimate the distributional air quality impacts of a carbon policy in 2030 (“Cap 50%”, scenario 3) relative to no carbon policy in 2030 (“Baseline”, scenario 2) and the historical year 2017 (“Hist.”, scenario 1). To ensure our primary policy case is a realistic projection, we draw from previous work that considered the potential impacts of a carbon pricing policy that reduces economy-wide emission in 2030 by 50% relative to the 2005 level. This scenario uses the outputs from an energy economic model of the U.S. economy combined with a power sector capacity expansion model (USREP-ReEDS) for a policy design that considers a range of technology cost assumptions and alternative emission allowance allocation

schemes, detailed in the Methods section and Yuan et al ²³. We then use a sensitivity scenario (scenario 4) to quantify the potential range for exposure reduction and equity outcomes for this particular policy due to uncertainty in the spatial distribution of sources, providing an upper and lower range for nationally averaged equity outcomes for each racial/ethnic group. We then conduct four different optimization scenarios (scenarios 5-8) around the primary policy case that test whether alternative emissions distributions that minimize POC mortalities under different constraints can better mitigate PM_{2.5} disparities while achieving the same 50% CO₂ reductions, and conduct two additional optimization scenarios (scenarios 9-10) testing the potential for mitigating disparities under 40% and 60% CO₂ reduction goals.

In the results I think there needs to be a table, because it has a lot of moving parts. I found “Potential for Disparity Mitigation through Alternative Carbon Reduction Distribution” section to be a bit confusing. There are so many options, and none of them are explained in detail.

Response: We appreciate the opportunity to clarify. We have now added a table detailing the different scenarios and policy cases included in our paper (also attached below).

Scenario number	Scenario name	Scenario category	Year	Policy case (CO ₂ targets)	Method for specifying emissions from individual sources
1	Hist. (2017)	Primary	2017	NA	historical data
2	Baseline (2030)	Primary	2030	NA	scaling based on energy economic model output
3	Cap 50% (2030)	Primary (policy)	2030	-50% relative to 2005	scaling based on energy economic model output
4	Uncertainty of Cap 50% (2030)	Sensitivity	2030	-50% relative to 2005	constraints by ReEDS region-sector but varying point source reductions
5	Nation-sector	Optimization	2030	-50% relative to 2005	minimizing POC mortalities with constraint on total sectoral emissions reductions
6	State-total	Optimization	2030	-50% relative to 2005	minimizing POC mortalities with constraint on total state-level emissions reductions

7	State-sector	Optimization	2030	-50% relative to 2005	minimizing POC mortalities with constraint on sectoral emissions reductions for each state
8	Nation-total	Optimization	2030	-50% relative to 2005	minimizing POC mortalities with no constraints
9	Nation-total (-40%)	Optimization	2030	-40% relative to 2005	minimizing POC mortalities with no constraints
10	Nation-total (-60%)	Optimization	2030	-60% relative to 2005	minimizing POC mortalities with no constraints

Table 1: Description of scenarios.

Also I feel like this paper has so few scenarios. It seems that the bulk of the work was done in the first paper (Yuan et al 2022) so I am wondering if the only add on here was the racial group analysis. There were many talked about in the political landscape. What about the renewable portfolio scenarios, or the low carbon standards? The 100% Renewable energy by 2035 or 2050.

Response:

We regret that our previous submission did not articulate that this work, while using the policy scenario in Yuan et al. 2022 as a starting point, used a very different method for analysis. We hope that Table 1 makes it clear the number of unique scenarios we conduct, and that the optimization cases are, in essence, the results of large ensembles.

Here, instead of focusing on specific policy proposals, we choose to evaluate a large set of emissions distribution scenarios that can achieve economy-wide emission reductions (~50%). While the main policy case is derived from a carbon pricing policy using an energy-economic model (Yuan et al. 2022), our main result relies on a large-scale optimization which essentially examines a huge set of potential scenarios that can achieve the emission reduction targets. We have now revised the paper both to add additional policy cases that evaluate how the air quality impacts might change with varying magnitudes of CO₂ emission reductions (representing 40 and 60% reductions, respectively), but also to emphasize that our results rely on a large-scale optimization. We also revised Figure 5 to include both the results from 5000 possible emission reduction scenarios (which all achieve the same CO₂ reduction as the “Cap 50%” scenario) as well as the several optimized scenarios (shown in the original Figure 5). We also expanded on our discussions of those optimized and non-optimized scenarios.

Figure 5: Impacts of alternative scenarios on POC exposure and disparities between POC and the overall population. All results are relative to the 2030 baseline scenario (Table 1, #2). The green X shows the results of our main policy case (Cap 50%, Table 1, #3). The colored dots show the optimization scenarios (with associated numbers from Table 1) which minimize the POC exposure under different constraints while achieving emissions reductions from 40% to 60% (emission targets are shown by colors). “Nation-total (max POC exposure)” shows a scenario that maximizes the POC mortality while achieving 50% CO₂ reductions, serving as a lower bound of the exposure reductions. The black dots show 5000 potential emission reduction scenarios, derived from random selection of sources and the amount of emissions reductions at each source, which all achieve the same level of CO₂ reduction (-50%). Those scenarios do not aim to optimize the POC mortalities but are used to show the full potential range of alternative policy scenarios.

Results section:

We explore whether selecting different sources of CO₂ reductions can better mitigate PM_{2.5} disparities while achieving the same total CO₂ reductions, using optimization for constructing emissions reduction scenarios around a set of policy cases (see Table 1 for scenario names and scenario numbers). As a primary policy case, we estimate the distributional air quality impacts of a carbon policy in 2030 (“Cap 50%”, scenario 3) relative to no carbon policy in 2030 (“Baseline”, scenario 2) and the historical year 2017 (“Hist.”, scenario 1). To ensure our primary policy case is a realistic projection, we draw from previous work that considered the potential impacts of a carbon pricing policy that reduces economy-wide emission in 2030 by 50% relative to the 2005 level. This scenario uses the outputs from an energy economic model of the U.S. economy combined with a power sector capacity expansion model (USREP-ReEDS) for a policy design that considers a

range of technology cost assumptions and alternative emission allowance allocation schemes, detailed in the Methods section and Yuan et al ²³. We then use a sensitivity scenario (scenario 4) to quantify the potential range for exposure reduction and equity outcomes for this particular policy due to uncertainty in the spatial distribution of sources, providing an upper and lower range for nationally averaged equity outcomes for each racial/ethnic group. We then conduct four different optimization scenarios (scenarios 5-8) around the primary policy case that test whether alternative emissions distributions that minimize POC mortalities under different constraints can better mitigate PM_{2.5} disparities while achieving the same 50% CO₂ reductions, and conduct two additional optimization scenarios (scenarios 9-10) testing the potential for mitigating disparities under 40% and 60% CO₂ reduction goals.

Potential for Disparity Mitigation through Alternative Carbon Reduction Distribution

The primary policy case described above illustrates the impact on disparities from regions and sectors that minimize CO₂ reduction cost (and associated spatial uncertainty). Next, we consider whether reducing the same amount of CO₂ from other combinations of regions and sectors could better mitigate air pollution exposure disparities, using an optimization approach. We conduct four different scenarios (Table 1, scenarios 5-8) to approximate different reduction strategies that might be achieved using either command-and-control or pricing mechanisms (see Methods/Optimization Approach). To do this, we use optimizations in which CO₂ reductions can come from different combinations of sources, minimizing PM_{2.5} associated mortality for POC, with a variety of different constraints. In all optimizations, we minimize POC mortality while keeping target CO₂ emissions reduction totals consistent under the different constraints and allowing individual sources to reduce in differing amounts to meet the overall target. Under “State-sector”, overall state and sectoral reductions are the same as in the “Cap 50%” policy, but the distribution of reductions among individual sources within each sector-state combination can vary. “State-total” maintains consistent reductions for each state, but allows reductions to come from different economic sectors. “Nation-sector” maintains Cap 50%’s distribution of sectoral reductions but allows reductions from those sectors to come from anywhere in the country. “Nation-total” sets a U.S.-wide cap and allows any source to reduce to meet it. The “State-sector” and “State-total” scenarios could correspond to efforts that states might introduce to prioritize CO₂ reductions in specific locations based on knowledge of sources that contribute the most to POC exposures. The least-constrained “Nation-total” scenario reflects a conceptual upper limit of the potential for targeting individual sources through national-scale policy design under a carbon reduction scenario of comparable magnitude. To test whether strengthening or weakening carbon reduction goals results in different outcomes, we also conduct “nation-total” scenarios for 40% and 60% reduction targets.

In Figure 5, we compare these optimization scenarios to the impacts estimated for the primary policy case, Cap 50%, plotting the change in average disparity between the total population and POC vs. the change in POC exposure. We show the different optimization

constraints, as well as 5,000 non-optimal scenarios for the 50% case which illustrates the range of potential outcomes from different source reduction choices. The 5,000 non-optimal scenarios are derived from random selection of sources and the amount of emissions reductions at each source in order to meet the same -50% CO₂ targets, without aiming to minimize POC mortalities. To illustrate the potential range of results, we also show a scenario that maximizes POC mortalities. We also show results from optimizations with national total 40% and 60% reductions.

While the primary policy scenario results in a widening of the air pollution exposure disparity for POC, further reductions in POC exposures are in principle achievable while still meeting the same CO₂ emissions reductions. The comparison between the Cap 50% scenario and the additional scenarios in which reductions can come from alternate sectors and regions implies that the least cost reduction opportunities identified by the carbon policy do not produce the greatest improvements in PM_{2.5} exposure. Prioritizing reductions in exposure for POC also reduces exposure for white people and the total population on average, suggesting a win-win of absolute gains from reducing sources that minimize POC mortality. However, this also means that the reduction in the overall disparity is limited, and substantial disparities remain. Allowing reductions to come from any source within a state can reduce disparities by 0.34% (the “State-total” scenario), while an additional 1.67% can be achieved by allowing reductions to come from different states (the “Nation-total” scenario). The sectoral contributions to this distribution are illustrated in Figure S4; the largest driver of additional reductions comes from the optimization constraint that allows for redistribution of emissions in the transportation sector, which is not substantially affected under the “Cap 50%” policy but largely responsible for the range of exposure reductions under the same CO₂ emission targets. Increasing the stringency of the carbon policy to 60% further reduces POC exposure, but also reduces exposure to the overall population, resulting in a smaller change in disparity relative to the baseline. A less ambitious (40%) carbon reduction, in contrast, results in a greater change in disparity, but less overall PM_{2.5} exposure benefit.

Why are some regions analyzed at the census tract level and others at the county level? It was unclear which is which.

Response: *We believe that we unintentionally gave this impression by an unclear clause in a sentence in the introduction describing our methodology. We have deleted the phrase “at county or census tract level” and clarified in our edits to the methods section that the spatial analysis is conducted at the scale of the underlying InMAP model output, and aggregated at state and national level.*

Fig 1 What type of electricity generation are you projecting because the PM looks very small but the SO_x is very high.

Response: Primary $PM_{2.5}$ emissions are smaller than SO_2 emissions in the 2017 (historical) scenario which reflects (roughly) the present-day electricity system in the US, and the order of magnitude difference in total emissions between them as measured in mass units (annual total SO_2 emissions are on the order of a million tonnes, while primary $PM_{2.5}$ totals around 100,000). This is a characteristic of the current energy mix and pollution control. SO_2 emissions from electricity drop substantially in the future scenarios as the carbon pricing scenario identifies the coal electricity sector as one of the most “efficient” sectors to decarbonize due to its high CO_2 intensity but relatively lower abatement cost.

I think in the results before you show figures there needs to be some discussion of how the industries are changing in each of your scenarios. Right now there is no context for how to interpret these numbers.

Response: We have moved the sentence describing the sectoral changes to the beginning of the section “Distributional Air Quality Impact of Carbon Policy,” before introduction of figure 1. This sentence reads:

Under the primary policy case, illustrating a carbon pricing policy, CO_2 emission reductions relative to Baseline in 2030 are driven mostly by the electricity sector (77%), followed by transportation (10%), industry (7%), and residential and commercial sectors (6%).

For a stylistic comment, the words at top were hard to read and connect at first (same stylistic comment for fig 3). I suggest moving them to the bottom of the x axis so the reader can more easily see the emissions you are referring to.

Response: Thank you for this comment. Following the reviewer’s suggestion, we have now edited the labels of figures 1 and 3 to make them more visible. The updated figures are attached here.

Figure 1. National emissions (billion metric tons (MT) for CO_2 and million MT for non- CO_2 pollutants) by pollutant and sector in Hist. (2017), Baseline (2030), and Cap 50% (2030). Values are displayed above each bar.

Figure 3. National population-weighted average PM_{2.5} exposure and relative disparity by race/ethnicity in Hist. (2017), Baseline (2030) and Cap 50% (2030). Disparity is calculated as the percentage difference between PM_{2.5} exposure for the given group and the total population.

Figure 2 in the main looks the same for panels e, h, and i. I think if this wants to stay on the main it should have different color bar or be a different figure. I think it doesn't make sense to

send people to the SI. Also for this to be in the main the authors should discuss the results from these ones. What can we infer from these graphs?

Response: Thank you for this comment. We have now moved panels e, h, and i to the SI figures, as these sectors do not contribute much emission reductions under the primary policy case illustrated in the figure. We have also revised the color bars for each sector to make the changes more visible.

Figure 2. Panels A - C: Annual average $\text{PM}_{2.5}$ concentrations ($\mu\text{g}/\text{m}^3$) under Cap 50% (2030) and changes relative to Baseline (2030) and Hist. (2017). Panels D to F: Change in concentrations under Cap 50% (2030) relative to Baseline (2030) from the three leading sectors. National population-weighted averages are listed under each respective title. Changes from the other sectors can be found in Figure S1.

Figure 4. I think this graphic does not convey your message well. If this is printed in black and white people will not be able to see the difference. I suggest adding in symbols. Also the title for racial groups should be moved to the X axis. It is hard to see the differences because there is overlap and the three symbols are all the same. I suggest changing to error bars or giving the different policies different symbols.

Response: Thank you for this suggestion. We have now changed the dots to error bars in figure 4, following the reviewer's suggestions.

Figure 4: Uncertainty range for the change in $PM_{2.5}$ exposures (red, panel A) and disparities (blue, panel B) by race/ethnicity between Cap 50% (2030) and Baseline (2030). Results are based on sensitivity simulations in which total reductions remain constant for each economic region and sector, while the reductions among different point sources are allowed to vary. Disparity is calculated as the percentage difference between $PM_{2.5}$ exposure for the given group and the total population.

Why is your methods named “Online methods”?

Response: We have double-checked the instructions for authors and have now confirmed that Nature Communications does include a traditional Methods section (other Nature journals have Online Methods). We apologize for missing this difference among journals in our previous submission and have revised the manuscript accordingly.

An equity outcome is not defined. I think this paper needs to state what an equity outcome for the racial/ethnic groups means in plain English. Also I think this paper relies so heavily on the two previously published papers that it makes this one hard to read. For example in the carbon pricing policy the CO₂ pricing policy (i.e., what level is the price), was not mentioned. I suggest a table with the relevant information about your scenarios, and potentially adding more scenarios.

Response: Thank you for this suggestion. We have now explicitly defined “disparities” and clarified in the paper that we use the “changes in pollution disparities” as the main equity outcome. As mentioned above, we have also included a table to detail the emission scenarios used in the paper.

The carbon price itself which was used to construct the policy base case is now included in the paper, but as this is an optimization of source reductions, the scenarios do not include carbon prices. We hope that the revision makes this clearer.

In the Baseline scenario (“Baseline”), results are calibrated to the Energy Information Administration’s Annual Energy Outlook 2020 reference case and in addition, reflect NREL’s Annual Technology Baseline 2019 Mid-Range electricity technology costs and performance characteristics, updated state clean energy policies, and a COVID-19 pandemic adjustment. The policy scenario (“Cap 50%”) imposes on the Baseline a national CO₂ cap-and-trade program that covers energy and industry-related CO₂ emissions and allows national trading of emissions allowances (at a price of \$14 in 2025 and rising to \$99 in 2030) without offsets or banking or borrowing across years. The scenario assumes that CO₂ emission allowances are distributed to states on a per-capita basis and that the state revenue raised from allowance sales are rebated to households on a per-capita basis. While other choices of allowance allocation schemes are evaluated by Yuan et al. (2022) affected economic welfare outcomes, they have negligible impact on emissions outcomes and therefore are not analyzed here.

Need more information on how you downscale from ReEDS regions which are very large to the census tract level.

Response: *We have clarified in the section under methods “Emissions Scaling Methodology” that the emissions locations are provided by the NEI, and that the scaling is applied uniformly to all sources within the regions. We have further clarified that our sensitivity analysis (scenario 4) is designed to test the uncertainty resulting from spatial heterogeneity. The revised text reads:*

For our base case, we apply a uniform scaling factor to all emission sources from a specific sector within each USREP-ReEDS region (which have locations specified by the NEI). To address the spatial uncertainty of estimated emissions reductions under the uncertainty scenario (Table 1, #4), we produce alternative emissions distributions that are consistent with CO₂ emissions reductions in the energy-economic modeling but allow point source emissions to vary within each USREP-ReEDS region for each sector.

I am not sure why in the methods the S scaling factor changes for each pollutant. Also in the methods the authors appear to contradict themselves. “while in the uniform scaling method S_i is uniform across all emission sources within a region and scaling variable set, here S_i is unique to each emissions source i as determined by the optimization.” I suggest reframing this sentence. Also if you have change this to have S not be uniform then this is no longer the uniform scaling method.

Response: *Thank you for this suggestion. The original language was indeed a bit confusing. We have edited the text in the method section for better clarity. To help address this confusion, we now only allow sources to reduce, rather than allowing some sources to increase while others reduce more. This is more realistic in terms of a carbon*

reduction policy, and allows us to explain the scaling factor in a more straightforward way.

The equation should be given in “Optimization Approach to Assess Alternative Carbon Reduction Distribution” for the objective function and constraints.

Response: Thank you for this suggestion. We have now substantially revised the optimization section, with a detailed description of the optimization approach, by including the objective function and constraints.

In the method section:

Optimization Approach to Assess Alternative Carbon Reduction Distribution

To explore the impacts of alternative scenarios that achieve the same level of CO₂ reductions on pollution exposure and disparities, we design an optimization approach to explore if emissions distributions that are different than those under the modeled carbon policy better mitigate national-scale air quality disparities while still achieving the same total CO₂ emissions reductions. To do this, we apply the following optimization methodology to minimize POC mortality while keeping CO₂ constant for respective emissions group combinations: “State-sector”, “State-total”, “National-sector”, “National-total.” The sectors considered here are electricity, transportation, industry, and residential sectors.

First, using the ISRM, we calculate marginal mortality values (total U.S. mortality caused per ton of emissions of primary PM_{2.5}, SO₂, NO_x, NH₃, and VOC) for emissions from each grid cell for POC, using the concentration response function from krewski et al.⁴⁰ and all-cause mortality incidence rates for the total population. By matching emissions to their respective marginal mortality values, we can then calculate the mortality caused by each source and pollutant.

We conduct the optimization method following the following equation. In the optimization approach, emissions that are eligible to vary are sources that (1) cause PM_{2.5}-related mortality; and (2) have non-zero CO₂ emissions in the 2030 baseline.

Maximize or minimize:

$$\text{objective function} = \sum_i S_i TM_i$$

Subject to:

$$\sum_i S_i CO_{2i} = \sum_i CO_{2i}$$

1. Total CO₂ emission reductions are fixed at a constant level (40 to 60% reductions relative to the 2005 level).
2. Emissions of any pollutant cannot be less than 0 (lower bound) and cannot be higher than the level in the 2030 baseline scenario (upper bound).

where:

- i = unique index of eligible emissions sources.
- TM_i = total mortality (for POC) caused by emissions at source i in the 2030 baseline.
- S_i = scaling factor (decision variable) applied to emissions of all pollutants at source i , allowed to range between 0 and 1. A source is thus allowed to be completely shut down emitting zero emissions (i.e. $S=0$), or emit as much as the baseline emission (i.e. $S=1$).

I think there should be more discussion of the different sectors and their impacts on vulnerable communities in the authors results section. Which sector has the biggest impact on disparities.

Response: We have included this discussion in the response to the next comment below, in the context of previous literature which addresses different sectors.

I would like to bring this recent paper in Nature communications, and another paper which appears similar, but a big difference is these paper below focus solely on the electricity sector. I think the authors should compare their results.

- Goforth, T., Nock, D. Air pollution disparities and equality assessments of US national decarbonization strategies. Nat Commun 13, 7488 (2022). <https://doi.org/10.1038/s41467-022-35098-4>
- Mayfield, E. N. Phasing out coal power plants based on cumulative air pollution impact and equity objectives in net zero energy system transitions. Environ. Res. Infrastruct. Sustain. 2, 021004 (2022). <https://iopscience.iop.org/article/10.1088/2634-4505/ac70f6>

Response: Thank you for this suggestion. Following the reviewer's suggestion, we have now provided an extended discussion comparing our results to the many recently published work on this topic including the two papers mentioned here, which address the sectoral impact on disparities. As the geographical region, sectors, and policy contexts are different, we discussed the similarity and differences of the high-level insights from each paper. Generally, we find that the two key insights we obtain from our analysis are broadly consistent with what the other papers found. First, we find that different decarbonization pathways can have different impacts on the existing pollution disparities, both in terms of sign and magnitude. This is consistent with findings from Goforth and Nock, which finds that decarbonizing electricity can exacerbate the pollution disparities without accounting for the air quality impacts of low-carbon technologies (such as bioenergy). Second, we find that maximizing overall air quality benefits (even for disadvantaged communities) does not always help reduce the pollution disparities between different populations. This also seems consistent with findings from Mayfield (2022) who studies the different pathways of phasing out coal power plants, Gallagher and Holloway (2022) who study the full decarbonization of electricity, industrial, and light-duty vehicles sectors, and Zhu et al who study the electrification strategies of building and trucks in California.

Our analysis contributes to a broader and emerging literature that documents the complex interactions between climate policies, overall air quality benefits, and pollution disparities. Our research reveals two insights about this complex interaction: first, we find that different decarbonization pathways can have different impacts on existing pollution disparities, both in terms of sign and magnitude. Second, we find that maximizing overall air quality benefits (even for disadvantaged communities) does not always help reduce the pollution disparities between different populations. Both insights are broadly consistent with previous papers that focus on different geographical regions, sectors, or policy contexts. For example, Mayfield 2022 studied the pathways to phase out coal power plants and found that the pathways that maximize overall health benefits do not always align with the pathways that mostly benefit the disadvantaged communities. Goforth and Nock, 2022 found that decarbonizing the electricity sector can exacerbate the pollution disparities without accounting for the air quality impacts of low-carbon technologies (such as bioenergy). These insights also hold beyond the electricity sectors (Gallagher and Holloway, 2022). For example, Zhu et al. evaluated two decarbonization strategies (building electrification and truck electrification) in California, and found that while building electrification generates greater overall air quality benefits, it is comparatively less beneficial to the disadvantaged communities. Building on this work, our optimization results further demonstrate that one likely reason for these insights is the tension between maximizing the overall air quality benefits and reducing pollution disparities for specific disadvantaged communities (as also suggested by Wang et al., 2022).

Reviewer #3:

This is an outstanding paper with important conclusions. The methods are state-of-the art and appropriate for the questions at hand. The results are of compelling importance for reaching the stated aims of the US Government to reduce air pollution disparities as a cornerstone of its climate policy.

I recommend publication with the addition of some minor additional analyses and discussion.

Response: *Thank you! We appreciate the positive assessment of our work.*

- A comment. I found one aspect of Figure 1 quite remarkable -- the reductions in CO₂ from this climate policy greatly outpace the reductions in the emissions of all of the other pollutants. And it's not so surprising -- it's well known that non-energy-related emissions play a major role in present-day exposures (see for example the flow diagrams of Thakrar et al ES&T Letters 2020). Among those pollutants that do have emissions reductions, it's also interesting to note that only NO_x and SO₂ really meaningfully reduce (and less than CO₂). I can see why this might be the case for a suite of policies highly focused on transport and electric power generation. So, my suggestion:

You might speak to the degree to which the results of your analysis are strongly driven by the emissions aspects of the story. Do we see limited disparity benefits from this simulated policy largely because it is considerably more effective at reducing CO₂ than emissions of PM precursors?

Response: *Thank you for the insightful comment. The reviewer is correct that the simulated carbon policy (cap 50%) has very limited effects because the scenario is more effective at reducing CO₂ than other PM_{2.5} precursors. This is now more clearly shown by our analysis of the alternative scenarios that achieve the same level of CO₂ reduction (see Figure 5). As shown in figure 5, the simulated carbon pricing scenario achieves relatively smaller impacts on closing the disparities between POC and the total population, among the potential range of the alternative scenarios. However, even the "best" scenario for reducing pollution exposure disparities (among all the potential alternative scenarios) can only marginally reduce the exposure disparities (reduce the disparity gap by <2.7% percentage points, compared to a disparity gap of 12%). Our analysis thus suggests that any scenario that aims to reduce CO₂ emissions by a similar magnitude is unlikely to substantially reduce the disparity gap of PM_{2.5}.*

We showed that a cap-and-trade policy instrument reduces exposure to PM_{2.5} for all racial/ethnic groups relative to Baseline, but does not substantially mitigate relative disparities in exposure. Black, Hispanic, and Asian people continue to experience disparities, while white people experienced less exposure than the total population on average. This is because the carbon policy achieves most reductions in the coal-fired electricity sector. Previous studies have showed that this sector disproportionately harms only Black and white people more than average (Tessum et al., 2021). In contrast, the

electricity sector contributes a relatively small fraction to population exposure overall, and key disparities arise from harder-to-decarbonize sectors with remaining emissions even under 50% cuts, such as industry and heavy-duty diesel transportation. These results are robust to assumptions about emissions reduction distribution, suggesting that the geographic distribution of source reductions under comparable policies do not drive substantial differences in outcomes with respect to disparities. As shown in our optimizations results (Figure 5), the simulated carbon pricing scenario actually achieves relatively smaller impacts on closing the disparities between POC and the total population, among the potential range of the alternative scenarios.

- Suggested supplementary analysis on spatial scales. Considerable recent work shows that PM2.5 disparities arise from multiple different spatial scales -- regional differences in demographics (e.g., the Southeastern US has higher PM and higher share of Black population); urban-rural differences (urban areas have higher PM and are more diverse) and finally the within-urban differences in exposure that arise due to forces of urban segregation and land use policy. All three spatial scales matter, of course, in contributing to inequalities. However, it is arguably the within-urban inequalities that often attract the largest attention, perhaps simply because of how readily apparent and ethically troubling they are. So, my suggestion:

You might consider a minor supplemental analysis to look at the degree to which your results differ when considered at different spatial scales. For example, you could do a within-urban sub-analysis by considering the distribution of disparity reduction in urban areas only by clipping to the InMAP grid cells that correspond to US census urbanized areas, and comparing the distribution of changes in each city. I don't think this needs more than an SI figure or two. PS - this comment was inspired by reading a new paper just published in ES&T Letters: Liu J and Marshall JD, DOI: 10.1021/acs.estlett.2c00826

Response: *Thank you for this great suggestion. Following the reviewer's comments, we have now performed a sensitivity analysis by just focusing on the urban areas. As suggested, we classify a grid cell as urban if it intersects with the US census urbanized areas (census data derived from 2017). We focused on the 20 most populous urban areas in the US (as defined by their populations in 2020). The 20 urban areas span 19421 InMAP grid cells (out of 52411 cells) and cover more than one-third of the US population (~110 million).*

The results of this sensitivity analysis are shown in Figure S3 and Table S1. Specifically, we estimate 1) the aggregated air pollution disparities of different racial/ethnic groups who live in the urbanized area (i.e. the results shown in Figure 4 but only limited to the population living in the 20 urban areas); 2) the within-city disparities for each of the 20 urban areas.

We find that the carbon policy has very limited impacts on the pollution disparities within each urban area. The pollution disparities between POC and average population actually increase by a small margin for most of the 20 areas, as a result of the carbon policy.

Furthermore, we find that the overall effects of policy on average disparities can fall outside the range of the city-level results (for example for the black population). This is because the simulated carbon policy achieves greater pollution exposure in cities with a higher percentage of the black population, even though the simulated carbon policies always disproportionately benefit non-black people more within each city.

Figure S3: Impacts of carbon policy on exposure disparities for each ethnic/racial group in the 20 major urbanized areas in the US. The box plot shows the range of changes in within-city disparity due to the simulated carbon policy scenario (cap 50% (2030)). The triangle shows the impacts on the average disparities pooling across the 20 urbanized areas (i.e. the estimates shown in Figure 3, but only for these 20 urbanized areas).

	Asian		Black		Hispanic		People of Color		White	
	baseline disparity	change in disparity	baseline in disparity	change in disparity	baseline in disparity	change in disparity	baseline in disparity	change in disparity	baseline in disparity	change in disparity
New York--Newark, NY--NJ--CT	9.5%	0.26%	10.0%	0.33%	12.0%	0.32%	10.7%	0.31%	-11.3%	-0.33%
Los Angeles--Long Beach--Anaheim, CA	-5.2%	-0.03%	9.0%	-0.03%	5.8%	0.03%	3.4%	0.01%	-8.3%	-0.02%
Chicago, IL--IN	2.2%	0.08%	12.4%	0.02%	7.7%	0.18%	8.5%	0.10%	-7.4%	-0.08%

Miami, FL	-8.3%	0.02%	1.8%	0.01%	14.4%	-0.01%	9.2%	0.00%	-17.0%	0.01%
Houston, TX	5.5%	-0.09%	1.2%	-0.06%	1.9%	0.01%	2.0%	-0.02%	-3.5%	0.04%
Dallas--Fort Worth--Arlington, TX	5.9%	0.22%	-1.8%	-0.10%	0.0%	0.01%	0.1%	0.00%	-0.1%	0.00%
Philadelphia, PA--NJ--DE--MD	5.3%	0.22%	15.6%	0.67%	11.5%	0.43%	12.4%	0.52%	-6.8%	-0.28%
Washington, DC--VA--MD	-4.1%	-0.48%	6.1%	0.81%	2.6%	0.05%	2.8%	0.31%	-3.4%	-0.37%
Atlanta, GA	3.0%	0.82%	3.6%	0.38%	2.6%	0.37%	3.2%	0.40%	-3.2%	-0.41%
Boston, MA--NH--RI	16.2%	0.14%	27.0%	0.17%	21.3%	0.20%	20.3%	0.16%	-6.8%	-0.06%
Phoenix--Mesa, AZ	0.2%	0.02%	4.3%	-0.03%	6.2%	-0.06%	5.1%	-0.04%	-3.5%	0.03%
Detroit, MI	-1.3%	-0.05%	10.3%	0.34%	4.8%	0.33%	7.9%	0.28%	-4.1%	-0.15%
Seattle, WA	13.7%	-0.01%	6.9%	0.43%	0.7%	0.02%	5.8%	0.09%	-2.8%	-0.04%
San Francisco--Oakland, CA	-1.4%	0.00%	15.3%	0.04%	6.9%	0.02%	3.9%	0.01%	-6.3%	-0.01%
San Diego, CA	3.5%	-0.04%	5.5%	0.02%	-1.6%	0.06%	0.4%	0.03%	-0.4%	-0.03%
Minneapolis--St. Paul, MN--WI	6.8%	0.12%	11.5%	0.17%	8.5%	0.15%	8.6%	0.14%	-2.7%	-0.04%
Tampa--St. Petersburg, FL	1.3%	0.06%	5.1%	0.25%	6.3%	0.27%	5.0%	0.22%	-2.5%	-0.11%
Denver--Aurora, CO	-0.9%	0.03%	16.7%	0.50%	6.7%	0.05%	7.0%	0.12%	-3.8%	-0.07%
Riverside--San Bernardino, CA	-3.3%	0.05%	-0.2%	-0.03%	6.4%	-0.02%	4.3%	-0.02%	-10.6%	0.04%
Baltimore, MD	-6.3%	-0.29%	7.3%	0.37%	2.4%	0.22%	4.9%	0.27%	-4.1%	-0.22%

Table S1: Impacts of carbon policy on exposure disparities for each ethnic/racial group within the 20 major urbanized areas in the US. The figure shows the existing pollution disparities within each city (the 2030 baseline scenario), and the changes in disparity due to the simulated carbon policy scenario (cap 50% (2030)).

Added text:

As air pollution exposure disparities arise at multiple scales – national, regional, and local (e.g., within urban areas), we further explore the impacts of the carbon policy on exposure disparities within urban areas and within each city. We find a considerable amount of heterogeneity at the state and urban area levels (see Figures S2 and S3). Figure S2 shows the change in disparities by state between Cap 50% and Baseline, showing large regional variation in impacts, driven by the correspondence between the population of each group and the location of largest reductions (as shown in Figure S1). While the policy narrows disparities in some states, widening disparities in other states mean that there is limited aggregate impact at national scale. At the local scale, we focus on the 20 most populous urban areas in the US. We find that the carbon policy actually exacerbates the within-city pollution disparities by a small margin, with large heterogeneity across different urban areas (Figure S3). We also find that the aggregated impacts of policy on exposure disparities are largely driven by influencing the exposure disparities at the regional level (instead of at the local scale). For example, while we find the policies exacerbate within-city exposure disparities for the black population in almost all 20 major urban areas, the policy actually helps reduce the exposure disparities when aggregated; this is because the regions with higher percentages of black population generally experience a larger reduction (despite the smaller reduction within each urban area.)

- Suggested addition to your discussion/conclusions: I find that this paper adds to a growing body of evidence that systemic racial-ethnic inequalities in air pollution exposure are unlikely to be resolved simply as a consequence of air pollutant emissions reductions. One relevant recent paper on this topic you might be interested in is Wang et al, PNAS 2022 (10.1073/pnas.2205548119). I think a reasonable working hypothesis for this broadly consistent result is that emissions sources are disparately concentrated near communities of color, and thus an economy-wide reduction of emissions does not eliminate relative disparities. So, my suggestion:

I think your conclusion section could state even more strongly your key result. In plain terms, expecting that climate policy will naturally lead to major reductions in relative racial/ethnic PM2.5 disparities runs contrary to the available evidence, and your valuable simulation provides a compelling demonstration of this point.

Response: *In the final paragraph of the discussion, we have added a sentence summarizing our result in plain terms. The text now reads:*

In summary, we show that simply reducing CO₂ sources over the next decade with a magnitude comparable to the current U.S. federal policy target, even if those sources are carefully selected, will not result in major reductions in air pollution exposure disparities among racial/ethnic groups in the U.S.

- Comment: I think your research very nicely tees up the questions of (i) why do these results arise, and (ii) how might we better design decarbonization policies to reduce disparities, or (iii)

perhaps these really need to be pursued in parallel with distinct approaches? These questions are very timely for current activities in US EPA and in the Biden Administration.

Response: Thank you for this great suggestion. Following the reviewer's comments, we have now expanded our discussion to clearly document the two broad insights we derive from our analysis, and discussed these insights in a comparison with the recently published work (including Wang et al., PNAS, suggested by the reviewer).

Our analysis contributes to a broader and emerging literature that documents the complex interactions between climate policies, overall air quality benefits, and pollution disparities. Our research reveals two insights about this complex interaction: first, we find that different decarbonization pathways can have different impacts on existing pollution disparities, both in terms of sign and magnitude. Second, we find that maximizing overall air quality benefits (even for the disadvantaged communities) does not always help reduce the pollution disparities between different populations. Both insights are broadly consistent with previous papers that focus on different geographical regions, sectors, or policy contexts. For example, Mayfield 2022 studied the pathways to phase out coal power plants and found that the pathways that maximize overall health benefits do not always align with the pathways that mostly benefit the disadvantaged communities. Goforth and Nock, 2022 found that decarbonizing the electricity sector can exacerbate the pollution disparities without accounting for the air quality impacts of low-carbon technologies (such as bioenergy). These insights also hold beyond the electricity sectors (Gallagher and Holloway, 2022). For example, Zhu et al., evaluated two decarbonization strategies (building electrification and truck electrification) in California, and found that while building electrification generates greater overall air quality benefits, it is comparatively less beneficial to the disadvantaged communities. Building on this work, our optimization results further demonstrate that one likely reason for these insights is the tension between maximizing the overall air quality benefits and reducing pollution disparities for specific disadvantaged communities (as also suggested by Wang et al., 2022).

REVIEWERS' COMMENTS

Reviewer #1 (Remarks to the Author):

In general the authors have done a fantastic job of addressing my comments. The paper is high quality, and I have a few minor suggestions below.

In the comparison section the comparison would benefit from specifics. For example "For example, Zhu et al. evaluated two decarbonization strategies (building electrification and truck electrification) in California, and found that while building electrification generates greater overall air quality benefits, it is comparatively less beneficial to the disadvantaged communities."

How much is "comparatively less"? I suggest adding numbers to this section to help with the comparisons.

These sentences are misleading "For example, Mayfield 2022 studied the pathways to phase out coal power plants and found that the pathways that maximize overall health benefits do not always align with the pathways that mostly benefit the disadvantaged communities. Goforth and Nock, 2022 found that decarbonizing the electricity sector can exacerbate the pollution disparities without accounting for the air quality impacts of low-carbon technologies (such as bioenergy)."

I find these sentences too high level to facilitate a comparison. Also, in its current form it sounds like you are saying the Goforth and Nock paper did not include low carbon tech. The Goforth and Nock paper included Bio power through the ReEDS model. They also use other low-carbon technologies see Figure 1. Nuclear is also considered a low carbon technology which is in their model. This sentence needs more explanation or rewording.

Suggested rewording

"For example, Mayfield 2022 studied the pathways to phase out coal power plants and found that the pathways that maximize overall health benefits (i.e. state a specific pathway here) do not always align with the pathways that mostly benefit the disadvantaged communities (i.e. state a specific pathway here). Goforth and Nock, 2022 found that without strict renewable energy and low carbon targets some electricity sector decarbonization pathways can exacerbate the pollution disparities."

Reviewer #3 (Remarks to the Author):

I reviewed the initial submission of this manuscript, and found it to be excellent. I suggested acceptance after a small number of generally minor revisions. On reading this revised submission, I appreciated the thoughtful and careful consideration of the review comments from the full set of reviewers. I believe that these revisions have only enhanced this excellent manuscript. I recommend publication as-is.

Response to Reviewers

Response: *We have made changes below to address the comments of the reviewers, specifically in referring to previous work. We have also added text noting the recently-published study of Polonik et al. (2023, PNAS) which was published while our paper was under review. We have also made editorial changes requested in the Nature Communications instructions to authors.*

REVIEWERS' COMMENTS

Reviewer #1 (Remarks to the Author):

In general the authors have done a fantastic job of addressing my comments. The paper is high quality, and I have a few minor suggestions below.

Response: *We appreciate the positive assessment of our work. We thank the reviewers for the excellent suggestions that helped improve our paper.*

In the comparison section the comparison would benefit from specifics. For example “For example, Zhu et al. evaluated two decarbonization strategies (building electrification and truck electrification) in California, and found that while building electrification generates greater overall air quality benefits, it is comparatively less beneficial to the disadvantaged communities.”

How much is “comparatively less”? I suggest adding numbers to this section to help with the comparisons.

Response: *Thank you for this suggestion. We have now added numbers when possible, to make the comparison more concrete. However, the calculation of scenario impacts on disadvantaged communities are complex (their estimates are based on a pre-defined equity index), and it would be difficult to interpret without the context. We are concerned that it might confuse the audience if we cite their numbers without a full explanation (which would be impossible given the space constraints). Nevertheless, we now included the quantitative difference of the total benefits across the two scenarios for more context of their paper.*

Revised text:

For example, Zhu et al. evaluated two decarbonization strategies (building electrification and truck electrification) in California, and found that while building electrification generates greater overall air quality benefits (~15%), it is comparatively less beneficial to disadvantaged communities²⁰.

These sentence are misleading “For example, Mayfield 2022 studied the pathways to phase out coal power plants and found that the pathways that maximize overall health benefits do not always align with the pathways that mostly benefit the disadvantaged communities. Goforth

and Nock, 2022 found that decarbonizing the electricity sector can exacerbate the pollution disparities without accounting for the air quality impacts of low-carbon technologies (such as bioenergy).”

I find these sentences too high level to facilitate a comparison. Also, in its current form it sounds like you are saying the Goforth and Nock paper did not include low carbon tech. The Goforth and Nock paper included Bio power through the ReEDS model. They also use other low-carbon technologies see Figure 1. Nuclear is also considered a low carbon technology which is in their model. This sentence needs more explanation or rewording.

Suggested rewording

“For example, Mayfield 2022 studied the pathways to phase out coal power plants and found that the pathways that maximize overall health benefits (i.e. state a specific pathway here) do not always align with the pathways that mostly benefit the disadvantaged communities (i.e. state a specific pathway here). Goforth and Nock, 2022 found that without strict renewable energy and low carbon targets some electricity sector decarbonization pathways can exacerbate the pollution disparities.”

***Response:** Thank you for this suggestion. We have reworded these sentences following the reviewer’s comment.*

Revised text:

For example, Mayfield studied the pathways to phase out coal power plants and found that the phase-out pathway that minimizes the cumulative mortalities from the electricity sectors is often not the highest-ranked pathway in terms of its impacts in reducing the pollution inequities measured using a suite of air quality equity indices the paper considered. Goforth and Nock found that without strict renewable energy and low carbon targets some electricity sector decarbonization pathways can exacerbate the pollution disparities.

Reviewer #3 (Remarks to the Author):

I reviewed the initial submission of this manuscript, and found it to be excellent. I suggested acceptance after a small number of generally minor revisions. On reading this revised submission, I appreciated the thoughtful and careful consideration of the review comments from the full set of reviewers. I believe that these revisions have only enhanced this excellent manuscript. I recommend publication as-is.

***Response:** Thank you! We appreciate the positive assessment of our work.*